CERN-TH-2024-147

# R-matrices and Miura operators in 5d Chern-Simons theory

**Nafiz Ishtiaque**$^a$ , **Saebyeok Jeong**$^b$ , and **Yehao Zhou**$^c$

$^a$*Institut des Hautes Études Scientifiques,*
*91440 Bures-sur-Yvette, France*

$^b$*Department of Theoretical Physics, CERN,*
*1211 Geneva 23, Switzerland*

$^c$*Kavli Institute for the Physics and Mathematics of the Universe (WPI), University of Tokyo,*
*Kashiwa, Chiba 277-8583, Japan*

*E-mail:* ishtiaque@ihes.fr, saebyeok.jeong@cern.ch, yehao.zhou@ipmu.jp

ABSTRACT: We derive Miura operators for $W$- and $Y$-algebras from first principles as the expectation value of the intersection between a topological line defect and a holomorphic surface defect in 5-dimensional non-commutative $\mathfrak{gl}(1)$ Chern-Simons theory. The expectation value, viewed as the transition amplitude for states in the defect theories forming representations of the affine Yangian of $\mathfrak{gl}(1)$, satisfies the Yang-Baxter equation and is thus interpreted as an R-matrix. To achieve this, we identify the representations associated with the line and surface defects by calculating the operator product expansions (OPEs) of local operators on the defects, as conditions that anomalous Feynman diagrams cancel each other. We then evaluate the expectation value of the defect intersection using Feynman diagrams. When the line and surface defects are specified, we demonstrate that the expectation value precisely matches the Miura operators and their products.

# 1 Introduction

The free-field realization of the $\mathcal{W}_N$-algebra [1, 2], a generalization of the Virasoro vertex algebra with higher-spin currents, is constructed using simple building blocks known as Miura operators.[1] These Miura operators are formal differential operators $R_i = \varepsilon_1 \partial_z - \varepsilon_2 \varepsilon_3 J_i(z)$, $i = 1, 2, \cdots, N$, from which the $\mathcal{W}_N$-algebra is generated through their product:

$$(\varepsilon_1 \partial_z - \varepsilon_2 \varepsilon_3 J_1(z))(\varepsilon_1 \partial_z - \varepsilon_2 \varepsilon_3 J_2(z)) \cdots (\varepsilon_1 \partial_z - \varepsilon_2 \varepsilon_3 J_N(z)) = \sum_{j=0}^{N} U_j(z)(\varepsilon_1 \partial_z)^{N-j}, \quad (1.1)$$

where $U_j(z)$ is the spin-$j$ generating current of the $\mathcal{W}_N$-algebra ($U_0(z) = 1$). Here, $J_i(z)$ is the current of the $i$-th $\widehat{\mathfrak{gl}}(1)$ vertex algebra with the operator product expansion (OPE) $J_i(z)J_j(w) \sim -\frac{1}{\varepsilon_2 \varepsilon_3} \frac{1}{(z-w)^2} \delta_{i,j}$, and $(\varepsilon_1, \varepsilon_2, \varepsilon_3) \in \mathbb{C}^3$, constrained by $\varepsilon_1 + \varepsilon_2 + \varepsilon_3 = 0$, parametrize the central charge of the $\mathcal{W}_N$-algebra by $c = N \left( 1 + (N^2 - 1) \left( \frac{\varepsilon_2}{\varepsilon_3} + \frac{\varepsilon_3}{\varepsilon_2} + 2 \right) \right)$.

In the above Miura transformation, we may consider exchanging the ordering of any two adjacent Miura operators in the product, from which an isomorphic $\mathcal{W}_N$-algebra should be generated. The isomorphism is realized by the Maulik-Okounkov R-matrix, which we schematically denote by $R_{\mathrm{MO}} \in \widehat{\mathfrak{gl}}(1) \widehat{\otimes} \widehat{\mathfrak{gl}}(1)$, satisfying [9]

$$R_i R_{i+1} R_{\mathrm{MO}} = R_{\mathrm{MO}} R_{i+1} R_i. \quad (1.2)$$

This relation is reminiscent of the Yang-Baxter equation satisfied by R-matrices, which plays the central role in the representation theory of quantum algebra. To make the connection concrete, it is required to interpret the Miura operators as R-matrices for representations of a certain quantum algebra.

Embedding the problem in the string/M-theory setting provides useful insights in this regard. It is well-known that the $\mathcal{W}_N$-algebra is the vertex algebra of local operators in a six-dimensional $\mathcal{N} = (2, 0)$ superconformal theory, subject to the $\Omega$-background [10–13]. This theory is realized as the low-energy effective theory on the worldvolume of $N$ parallel M5-branes in the M-theory, also subject to the $\Omega$-background [14]. Due to the $\Omega$-background, this *twisted* M-theory is localized into a 5-dimensional holomorphic-topological non-commutative $\mathfrak{gl}(1)$ Chern-Simons theory, where the $N$ parallel M5-branes descend to a holomorphic surface defect. The $\mathcal{W}_N$-algebra is then viewed as the vertex algebra of local operators on this surface

---

[1]In the present work, $\mathcal{W}_N$-algebra always refers to the $\mathcal{W}$-algebra of $\mathfrak{gl}(N)$; the direct sum of the usual $\mathcal{W}$-algebra of $\mathfrak{sl}(N)$ and a decoupled free boson. In the $N \to \infty$ limit, the $\mathcal{W}_N$-algebra becomes the $\mathcal{W}_\infty$-algebra [3–8]. What we call $\mathcal{W}_\infty$-algebra in this paper is traditionally denoted by $\mathcal{W}_{1+\infty}$ in the literature, where "1+" refers to the additional $\widehat{\mathfrak{gl}}(1)$ current for the free boson. We always include the free boson throughout this work, and we omit "1+" to simplify the notation.

defect. In particular, in the aforementioned free-field realization of the $\mathcal{W}_N$-algebra, each copy of the $\widehat{\mathfrak{gl}}(1)$ vertex algebra corresponds to one of the $N$ M5-branes comprising the surface defect.

It was subsequently postulated that the Miura operators could also be constructed within the twisted M-theory framework by inserting an M2-brane that transversally intersects these $N$ parallel M5-branes [15]. The intersection point was shown to support a fermionic zero mode, whose expectation value was proposed to precisely correspond to the Miura operator. The construction was also generalized to include non-parallel M5-branes with varying orientations. The intersection of an M2-brane with a non-transversally intersecting M5-brane was suggested to produce a *pseudo-differential* Miura operator, which, together with the usual Miura operators, provides a free-field realization of the $Y$-algebra [16, 17]. Recently, the Miura transformation of the $q$-deformed $W$- and $Y$-algebras was also established from the M2-M5 intersections in the multiplicative uplift of the M-theory background [18]. See also [19] for related work on the intersections of M2-branes and M5-branes.

In the 5d non-commutative $\mathfrak{gl}(1)$ Chern-Simons theory, the newly introduced M2-brane engineers a topological line defect. The gauge-invariance of the coupling of the line defect and the surface defect imposes strict constraints on the OPEs of local operators on these respective defects, requiring that the algebra of local operators on the former and the mode algebra of local operators on the latter be representations of the affine Yangian of $\mathfrak{gl}(1)$, $Y(\widehat{\mathfrak{gl}}(1))$ [20]. The M2-M5 intersection then passes to a gauge-invariant intersection of a line defect and a surface defect, which supports a space of local operators built from the (completed) tensor product of the two representations of $Y(\widehat{\mathfrak{gl}}(1))$. When the line defect intersects with two surface defects, re-ordering the two surface defects in the topological direction should yield an equivalent defect configuration up to the conjugation of the transition matrix between the two surface defects. The local operator at the intersection of a line defect and a surface defect is thus interpreted as an R-matrix of the affine Yangian of $\mathfrak{gl}(1)$.

However, it remains to be confirmed whether these R-matrices obtained from the M2-M5 intersection are indeed the Miura operators discussed above. This is the main goal of the present work. We aim to provide a first-principle derivation of the Miura operators in the perturbative study of the 5d non-commutative Chern-Simons theory. Note that the line defect and the surface defect interact with each other by exchanging the gauge field. The expectation value, or the transition amplitude for the states of the defect theories in time-radial ordering, can be evaluated using Feynman diagrams with increasing numbers of propagators and bulk interaction vertices connecting the two defects. A crucial factor in this perturbative analysis is the back-reaction of the M5-branes, which creates a singularity in the fields along the locus of the surface defect [20]. This back-reaction can be incorporated into the evaluation of the expectation value as a perturbative expansion in the number $N$ of M5-branes (see [21] for a study of holomorphic Chern-Simons theory, where the branes engineering surface defects also source a Beltrami differential by their back-reaction). We compute the expectation value for the intersection of a generic line defect and a generic surface defect, only with the orientation of the M5-branes fixed. Once the line defect and the surface defect are specified, we find that

the expectation value, mapped through the associated representations, precisely agrees with the Miura operators and their products.

The paper is organized as follows. In section 2, we begin by reviewing the 5-dimensional non-commutative $\mathfrak{gl}(1)$ Chern-Simons theory, with a focus on the Feynman rules for its perturbative analysis. We also introduce the topological line defect and the holomorphic surface defect, explaining how they couple to the 5d Chern-Simons theory. In section 3, we determine the algebra of local operators on the line defect and the surface defect by deriving their OPEs, based on the condition that anomalous Feynman diagrams cancel each other. In section 4, we demonstrate that the fusion of defects is governed by the coproduct structure, which exactly reproduces the known (meromorphic) coproduct of the 1-shifted affine Yangian of $\mathfrak{gl}(1)$ for the line defect and the $\mathcal{W}_\infty$-algebra for the surface defect. In section 5, we explain the Miura operators are the R-matrices for the representations of the affine Yangian of $\mathfrak{gl}(1)$ assigned to an M2-brane and an M5-brane. We compute the expectation value of the intersection of a generic line defect and a generic surface defect with a fixed orientation of the M5-branes. We show that, when the line defect and the surface defect are specified so that this expectation value is mapped through the associated representations, it exactly reproduces the Miura operators and their products. We conclude in section 6 with discussions. In the appendices, we review the definitions and relevant properties of the affine Yangian of $\mathfrak{gl}(1)$, the 1-shifted affine Yangian of $\mathfrak{gl}(1)$, and the $\mathcal{W}_\infty$-algebra. We also provide a proof of vanishing theorems for certain types of Feynman diagrams.

**Acknowledgement.** The authors thank Davide Gaiotto, Alba Grassi, Nathan Haouzi, Shota Komatsu, Jihwan Oh, Tomáš Procházka and Miroslav Rapčák for discussions and collaboration on related subjects. NI is supported at IHES by Huawei Young Talents fellowship. The work of SJ is supported by CERN and CKC fellowship. Kavli IPMU is supported by World Premier International Research Center Initiative (WPI), MEXT, Japan.

*Note added: The submission of the manuscript to arXiv was coordinated with* [22].

## 2 Five-dimensional Chern-Simons theory on $\mathbb{R} \times \mathbb{C}^2$

The twisted and $\Omega$-deformed M-theory on $\mathbb{R}^2_{\varepsilon_1} \times \mathbb{R}^2_{\varepsilon_2} \times \mathbb{R}^2_{\varepsilon_3} \times \mathbb{R}_t \times \mathbb{C}_x \times \mathbb{C}_z$, by compactification to IIA theory and localization, reduces to the 5-dimensional holomorphic-topological non-commutative $\mathfrak{gl}(1)$ Chern-Simons theory on $\mathbb{R}_t \times \mathbb{C}_x \times \mathbb{C}_z$ [14]. The $\Omega$-deformation parameters are required to satisfy a Calabi-Yau 3-fold condition

$$\varepsilon_1 + \varepsilon_2 + \varepsilon_3 = 0 \tag{2.1}$$

in order to preserve the supersymmetry for the localization. The resulting 5d CS theory is defined by the action

$$S = \frac{1}{\varepsilon_1} \int_{\mathbb{R} \times \mathbb{C}^2} \mathrm{d}x \wedge \mathrm{d}z \wedge \left( \frac{1}{2} A \star_\varepsilon \mathrm{d}A + \frac{1}{3} A \star_\varepsilon A \star_\varepsilon A \right), \tag{2.2}$$

where $A$ is a partial $\mathfrak{gl}(1)$-connection:

$$A = A_t \mathrm{d}t + A_{\bar{x}} \mathrm{d}\bar{x} + A_{\bar{z}} \mathrm{d}\bar{z}, \tag{2.3}$$

and $\star_\varepsilon$ is the Moyal product,

$$f \star_\varepsilon g = \sum_{n=0}^{\infty} \frac{\varepsilon^n}{2^n n!} \epsilon_{i_1 j_1} \cdots \epsilon_{i_n j_n} \left( \frac{\partial}{\partial z_{i_1}} \cdots \frac{\partial}{\partial z_{i_n}} f \right) \wedge \left( \frac{\partial}{\partial z_{j_1}} \cdots \frac{\partial}{\partial z_{j_n}} g \right). \tag{2.4}$$

Here, $\epsilon_{ij}$ is the antisymmetric symbol, and $(z_1, z_2) = (x, z)$. The Moyal product reflects the non-commutativity $[x, z] = \varepsilon$ of the holomorphic coordinates on $\mathbb{C}_x \times \mathbb{C}_z$. The parameter $\varepsilon$ is a function of the $\Omega$-deformation parameters, determined through the localization process. The localization of the twisted M-theory to the 5d Chern-Simons theory takes place in three steps [14]. First, the flat metric on $\mathbb{R}^2_{\varepsilon_2} \times \mathbb{R}^2_{\varepsilon_3}$ is continuously deformed to that of the single-centered Taub-NUT space, which is viewed as a circle bundle over $\mathbb{R}^3$. Then, the twisted M-theory gets compactified along the Taub-NUT circle. Second, the resulting IIA background is interpreted as the back-reaction of an emergent single D6-brane supported on $\mathbb{R}^2_{\varepsilon_1} \times \mathbb{R}_t \times \mathbb{C}_x \times \mathbb{C}_z$ in the presence of a constant B-field. Third, the 7-dimensional effective gauge theory on the worldvolume of the D6-brane is subject to an $\Omega$-deformation due to an $\varepsilon_1$-dependent RR 3-form, and is thus localized into the 5d non-commutative Chern-Simons theory.

In this process, the constant B-field is what leads to the non-commutativity (2.4) on the holomorphic planes. Starting from the M-theory background, we find $\varepsilon = 2\pi \left( \varepsilon_2 + \frac{\varepsilon_1}{2} \right)$ at the leading orders in $\frac{\varepsilon_1}{\varepsilon_2}$, though the effect of the $\varepsilon_1$-dependent RR 3-form in the back-reaction process is unclear at the moment. For now, we prefer to leave $\varepsilon$ as an unspecified function of the $\Omega$-background parameters and determine it later from the constraint of anomaly cancellation in the presence of defects in the CS theory. We shall find that $\varepsilon$ indeed coincides with $2\pi \left( \varepsilon_2 + \frac{\varepsilon_1}{2} \right)$ at the leading orders but receives *quantum* corrections, by which we refer to order-by-order corrections in powers of $\frac{\varepsilon_1}{\varepsilon_2}$.

The action (2.2) is invariant under the following gauge transformation:

$$A \mapsto A + (\mathrm{d}t\partial_t + \mathrm{d}\bar{x}\partial_{\bar{x}} + \mathrm{d}\bar{z}\partial_{\bar{z}})c + [A, c]_{\star_\varepsilon} \tag{2.5}$$

where $c$ is any complex-valued function on $\mathbb{R} \times \mathbb{C}^2$ and the commutator is defined using the Moyal product as

$$[A, c]_{\star_\varepsilon} := A \star_\varepsilon c - c \star_\varepsilon A = \varepsilon(\partial_x A \partial_z c - \partial_z A \partial_x c) + \mathcal{O}(\varepsilon^2). \tag{2.6}$$

Up to a total derivative, the 5d CS action is equivalent to

$$S = \frac{1}{\varepsilon_1} \int_{\mathbb{R} \times \mathbb{C}^2} \mathrm{d}x \wedge \mathrm{d}z \wedge \left( \frac{1}{2} A \wedge \mathrm{d}A + \frac{1}{3} A \wedge (A \star_\varepsilon A) \right). \tag{2.7}$$

## 2.1 Feynman rules for the bulk theory

The propagator of the theory (2.7) is a 2-form $P$, required to satisfy:

$$\frac{1}{\varepsilon_1} \mathrm{d}x \wedge \mathrm{d}z \wedge \mathrm{d}P(t, x, \bar{x}, z, \bar{z}) = \delta^{(5)}(t, x, \bar{x}, z, \bar{z}) \tag{2.8}$$

where the delta 5-form is normalized as:

$$\int_{\mathbb{R} \times \mathbb{C}^2} \delta^{(5)} = 1.$$
(2.9)

To find an explicit formula for the propagator we also need to choose a gauge fixing condition – we choose the analogue of the Lorenz gauge in the present holomorphic-topological setting:

$$(\partial_t \iota_{\partial_t} + 4\partial_x \iota_{\partial_{\bar{x}}} + 4\partial_z \iota_{\partial_{\bar{z}}}) P(t, x, \bar{x}, z, \bar{z}) = 0.$$
(2.10)

The solution to (2.8) and (2.10) is:

$$P(v) = \frac{3\varepsilon_1}{16\pi^2} \frac{\frac{1}{2} t d\bar{x} \wedge d\bar{z} + \bar{x} d\bar{z} \wedge dt - \bar{z} d\bar{x} \wedge dt}{(t^2 + |x|^2 + |z|^2)^{\frac{5}{2}}}.$$
(2.11)

Here and afterward, we use $v = (t, x, \bar{x}, z, \bar{z})$ to refer to all the coordinates of the space-time. In such cases we shall also use the notation $v^\mu$ for $\mu = 1, \cdots, 5$ to refer to individual coordinates, e.g., $v^1 = t, v^2 = x, v^3 = \bar{x}$, etc.

We can verify that (2.11) is indeed a solution to (2.8). It is easy to check that $dx \wedge dz \wedge dP = 0$ away from the origin. So $dx \wedge dz \wedge dP$ must be proportional to the delta function and the proportionality factor can be determined by integrating it over any volume containing the origin. We can take the ball of radius $r$ centered at the origin. At the surface of the ball we have

$$P(v)\Big|_{t^2 + |x|^2 + |z|^2 = r^2} = \frac{3\varepsilon_1}{16\pi^2 r^5} \left( \frac{1}{2} t d\bar{x} \wedge d\bar{z} + \bar{x} d\bar{z} \wedge dt - \bar{z} d\bar{x} \wedge dt \right)\Big|_{t^2 + |x|^2 + |z|^2 = r^2}.$$
(2.12)

Therefore, by the Stokes theorem, we also have the equality:

$$\begin{aligned}
&\frac{1}{\varepsilon_1} \int_{t^2 + |x|^2 + |z|^2 < r^2} dx \wedge dz \wedge dP \\
&= \frac{3}{16\pi^2 r^5} \int_{t^2 + |x|^2 + |z|^2 < r^2} dx \wedge dz \wedge d\left( \frac{1}{2} t d\bar{x} \wedge d\bar{z} + \bar{x} d\bar{z} \wedge dt - \bar{z} d\bar{x} \wedge dt \right) \\
&= -\frac{15}{32\pi^2 r^5} \int_{t^2 + |x|^2 + |z|^2 < r^2} dt \wedge dx \wedge d\bar{x} \wedge dz \wedge d\bar{z} \\
&= 1.
\end{aligned}$$
(2.13)

Note that our holomorphic forms are normalized so that $dx \wedge d\bar{x}$ is $-2i$ times the standard Euclidean volume form on $\mathbb{R}^2$ and similarly for $z, \bar{z}$.

By definition, the propagator is related to the 2-point correlation functions:

$$P(v_{12}) = \frac{1}{2} \langle A_\mu(v_1) A_\nu(v_2) \rangle dv_{12}^\mu dv_{12}^\nu$$
(2.14)

where $v_i = (t_i, x_i, \bar{x}_i, z_i, \bar{z}_i)$ for $i = 1, 2$ are two arbitrary points of the space-time and $v_{12} := v_1 - v_2$ refers to their difference. From (2.11) we can then extract the expressions for the

correlation functions:

$$\langle A_{\bar{x}}(v_1)A_{\bar{z}}(v_2)\rangle = \frac{3\varepsilon_1}{32\pi^2}\frac{t_{12}}{(t_{12}^2+|x_{12}|^2+|z_{12}|^2)^{\frac{5}{2}}},$$

$$\langle A_{\bar{z}}(v_1)A_t(v_2)\rangle = \frac{3\varepsilon_1}{16\pi^2}\frac{\bar{x}_{12}}{(t_{12}^2+|x_{12}|^2+|z_{12}|^2)^{\frac{5}{2}}}, \tag{2.15}$$

$$\langle A_{\bar{x}}(v_1)A_t(v_2)\rangle = \frac{3\varepsilon_1}{16\pi^2}\frac{-\bar{z}_{12}}{(t_{12}^2+|x_{12}|^2+|z_{12}|^2)^{\frac{5}{2}}}.$$

In Feynman diagrams, we shall denote these correlation functions by a wavy line:

$$v_1{}^{\mu}\wwwwww^{\nu}v_2 \;=\; \langle A_{\mu}(v_1)A_{\nu}(v_2)\rangle. \tag{2.16}$$

In practice, the points will be integrated over and the space-time indices properly contracted, and thus usually omitted from diagrams.

The 5d CS action (2.7) contains infinitely many interaction terms of increasingly higher order in $\varepsilon$, starting from a linear term. We shall restrict our attention to Feynman diagrams of no higher order than $\varepsilon^3$ and therefore we only need the first few interaction terms:

$$\frac{1}{3\varepsilon_1}A \wedge (A \star_{\varepsilon} A) = \frac{\varepsilon}{3\varepsilon_1}A \wedge \partial_x A \wedge \partial_z A$$
$$+ \frac{\varepsilon^3}{24\varepsilon_1}A \wedge \left(\frac{1}{3}\partial_x^3 A \wedge \partial_z^3 A - \partial_x^2\partial_z A \wedge \partial_z^2\partial_x A\right) + \mathcal{O}(\varepsilon^4). \tag{2.17}$$

It is crucial to note that the second-order term in the expansion of the Moyal product vanishes identically. In Feynman diagrams, we represent the two interaction terms above separately as:

$$\overset{\textstyle\diagup\!\!\!|\!\!\!\diagdown}{\textstyle(1)} \;=\; \frac{\varepsilon}{3\varepsilon_1}\int_{\mathbb{R}\times\mathbb{C}^2}\mathrm{d}x \wedge \mathrm{d}z \wedge A \wedge \partial_x A \wedge \partial_z A\,, \tag{2.18a}$$

$$\overset{\textstyle\diagup\!\!\!|\!\!\!\diagdown}{\textstyle(3)} \;=\; \frac{\varepsilon^3}{24\varepsilon_1}\int_{\mathbb{R}\times\mathbb{C}^2}\mathrm{d}x \wedge \mathrm{d}z \wedge A \wedge \left(\frac{1}{3}\partial_x^3 A \wedge \partial_z^3 A - \partial_x^2\partial_z A \wedge \partial_z^2\partial_x A\right). \tag{2.18b}$$

## 2.2 Topological line defect

Due to the holomorphic-topological nature of the 5d CS theory, topological line defects can only be supported on the line $\mathbb{R}_t$, located at specific positions on the holomorphic planes $\mathbb{C}_x \times \mathbb{C}_z$. Since the worldline is topological, the correlation function of local operators on the line defect does not depend on their local positions, but only on their ordering. Namely, the local operators form an associative algebra defined by their OPEs.

The coupling of the topological line defect to the 5d CS theory is built from a product of these local operators on the defect and the modes of the ghost at the locus of the line defect through the topological descent procedure. The result is the generalized Wilson line,

$$\mathrm{Pexp}\sum_{m,n=0}^{\infty}\alpha_l\int_{\mathbb{R}_t}\mathrm{d}t\,\frac{t_{m,n}}{m!n!}\partial_x^m\partial_z^n A_t, \tag{2.19}$$

inserted in the path integral, where $\{t_{m,n}\}_{m,n\in\mathbb{Z}_{\geq 0}}$ are the generators of the algebra of local operators on the line defect. The BRST invariance of the coupling requires the algebra of local operators to be a representation of a certain universal associative algebra, which is proven to be the 1-shifted affine Yangian of $\mathfrak{gl}(1)$ in the present case [14, 23]. We recall the definition and the relevant properties of the 1-shifted affine Yangian of $\mathfrak{gl}(1)$ in appendix B.

In the twisted M-theory origin, the topological line defects in the 5d CS theory descend from the M2-branes supported on $\mathbb{R}^2_{\varepsilon_c} \times \mathbb{R}_t$, where $c \in \{1, 2, 3\}$. The most generic line defect $\mathcal{L}_{l,m,n}$ is thus constructed by $(l, m, n)$ M2-branes wrapping $\mathbb{R}^2_{\varepsilon_1} \times \mathbb{R}_t$, $\mathbb{R}^2_{\varepsilon_2} \times \mathbb{R}_t$, and $\mathbb{R}^2_{\varepsilon_3} \times \mathbb{R}_t$, respectively. We denote the algebra of local operators on $\mathcal{L}_{l,m,n}$ by $\mathrm{M2}_{l,m,n}$, so that there exists a surjective homomorphism

$$\rho_{\mathrm{M2}_{l,m,n}} : Y_1(\widehat{\mathfrak{gl}}(1)) \twoheadrightarrow \mathrm{M2}_{l,m,n}. \tag{2.20}$$

These representations are identified to be the spherical rational double affine Hecke algebra and its generalizations [15, 24].

For later use, let us reproduce here the representations assigned to one or two parallel M2-branes. The representation $\rho_{\mathrm{M2}_{1,0,0}}$ is fully determined from

$$\rho_{\mathrm{M2}_{1,0,0}}(t_{0,n}) = \frac{1}{\varepsilon_1} z^n, \qquad \rho_{\mathrm{M2}_{1,0,0}}(t_{2,0}) = \varepsilon_1 \partial_z^2, \tag{2.21}$$

by using the commutation relations (B.1) and (B.2). The representations for single M2-brane with other two orientations can be obtained simply by permuting $(\varepsilon_1, \varepsilon_2, \varepsilon_3)$. Similarly, the representation $\rho_{\mathrm{M2}_{2,0,0}}$ is fully determined by

$$\rho_{\mathrm{M2}_{2,0,0}}(t_{0,n}) = \frac{1}{\varepsilon_1}(z_1^n + z_2^n), \qquad \rho_{\mathrm{M2}_{2,0,0}}(t_{2,0}) = \varepsilon_1(\partial_{z_1}^2 + \partial_{z_2}^2) + \frac{\varepsilon_2 \varepsilon_3}{\varepsilon_1} \frac{2}{(z_1 - z_2)^2}. \tag{2.22}$$

The representations for two parallel M2-branes with other two orientations can be obtained by permuting $(\varepsilon_1, \varepsilon_2, \varepsilon_3)$.

## 2.3 Holomorphic surface defect

In the 5d CS theory, the support of holomorphic surface defects can be $\mathbb{C}_x$-plane or the $\mathbb{C}_z$-plane. We will always fix the support to be $\mathbb{C}_z$. Since the worldvolume is holomorphic, the correlation function of local operators on the surface defect only has dependence on their holomorphic coordinates. The OPE of local operators can be singular only when these holomorphic coordinates collide, and thus defines a vertex algebra structure on the space of local operators.

The coupling of the surface defect to the 5d CS theory is realized by the coupling action,

$$\sum_{m=0}^{\infty} \alpha_s \int_{\mathbb{C}_z} \mathrm{d}^2 z \, \frac{1}{m!} W^{(m+1)} \partial_x^m A_{\bar{z}}, \tag{2.23}$$

where $W^{(m+1)}(z) = \sum_{n \in \mathbb{Z}} W_n^{(m+1)} z^{-m-n-1}$ is a spin-$(m+1)$ local operator. The BRST invariance of the coupling requires the vertex algebra of these local operators to form a representation of the $\mathcal{W}_\infty$-algebra [14].

In the twisted M-theory setting, the holomorphic surface defects are engineered by the M5-branes supported on $\mathbb{R}^2_{\varepsilon_{c+1}} \times \mathbb{R}^2_{\varepsilon_{c-1}} \times \mathbb{C}_z$, where $c \in \{1, 2, 3\}$. The most generic surface defect $\mathcal{S}_{L,M,N}$ supported on $\mathbb{C}_z$ is therefore constructed by $(L, M, N)$ M5-branes wrapping $\mathbb{R}^2_{\varepsilon_2} \times \mathbb{R}^2_{\varepsilon_3} \times \mathbb{C}_z$, $\mathbb{R}^2_{\varepsilon_1} \times \mathbb{R}^2_{\varepsilon_3} \times \mathbb{C}_z$, and $\mathbb{R}^2_{\varepsilon_1} \times \mathbb{R}^2_{\varepsilon_2} \times \mathbb{C}_z$, respectively. We denote the vertex algebra on the surface defect $\mathcal{S}_{L,M,N}$ by $\mathrm{M5}_{L,M,N}$, and its mode algebra by $U(\mathrm{M5}_{L,M,N})$. Accordingly, there is a surjective vertex algebra homomorphism

$$\rho_{\mathrm{M5}_{L,M,N}} : \mathcal{W}_\infty \twoheadrightarrow \mathrm{M5}_{L,M,N}. \tag{2.24}$$

The representation $\mathrm{M5}_{L,M,N}$ of $\mathcal{W}_\infty$ is expected to be the $Y_{L,M,N}$-algebra, originally constructed as the vertex algebra at the corner [16], see also related works in [25–28]. It is a generalization of the $\mathcal{W}$-algebra of $\mathfrak{gl}(N)$, $\mathcal{W}_N = Y_{N,0,0}$, realized on $N$ parallel M5-branes. In the present work, we will explicitly confirm that some of the OPEs of local operators on the surface defect $\mathcal{S}_{N,0,0}$ are indeed those of the $\mathcal{W}_N$-algebra.

The perturbative study of the 5d CS theory in the presence of the surface defect involves a subtlety which was absent for the line defect. Under the reduction to the IIA theory, the $N$ M5-branes supported on $\mathbb{R}^2_{\varepsilon_2} \times \mathbb{R}^2_{\varepsilon_3} \times \mathbb{C}_z$ reduce to $N$ D4-branes intersecting a single emergent D6-brane along the holomorphic plane $\mathbb{C}_z$. By quantizing D4-D6 open strings we obtain chiral fermions living on $\mathbb{C}_z$, which couple to both the 5d CS theory and the effective theory for the D4-D4 open string. Even though each coupling seemingly suffers from anomaly, the two contributions cancel each other so that the whole system is non-anomalous [29]. Now, the D4-D4 open string can be compensated by its back-reaction for the closed string background. In this perspective, the D4-branes dissolve into the background sourcing a singularity of the fields in the 5d CS theory along the support of the holomorphic surface defect (namely, $\mathbb{C}_z$-plane in our case), which guarantees the absence of the anomaly for the coupling of the chiral fermions [14].

In the presence of this back-reaction, the perturbation theory for the 5d CS theory is expanded around $A = A^{(0)}$, instead of around $A = 0$, satisfying

$$\partial F^{(0)} = 0, \qquad \frac{\mathrm{i}}{2\pi} \mathrm{d}F^{(0)} = \frac{\mathrm{i}}{2\pi} (\mathrm{d}t \partial_t + \mathrm{d}\bar{x} \partial_{\bar{x}}) F^{(0)} = N \frac{\varepsilon_1}{\varepsilon} \delta^3(t, x, \bar{x}), \tag{2.25}$$

where $F^{(0)} = \partial A^{(0)}$ [14]. The solution can be explicitly written as

$$F^{(0)} = \frac{N}{2} \frac{\varepsilon_1}{\varepsilon} \frac{\bar{x} \mathrm{d}t \wedge \mathrm{d}x + \frac{1}{2} t \mathrm{d}x \wedge \mathrm{d}\bar{x}}{(t^2 + |x|^2)^{\frac{3}{2}}}. \tag{2.26}$$

It is straightforward to see that (2.26) satisfies (2.25). In particular, as in the computation of (2.13), we have

$$\frac{\mathrm{i}}{2\pi} \int_{t^2 + |x|^2 \leq r} (\mathrm{d}t \partial_t + \mathrm{d}\bar{x} \partial_{\bar{x}}) F^{(0)} = \frac{\mathrm{i}}{2\pi} \frac{N}{2} \frac{\varepsilon_1}{\varepsilon} \frac{3}{2r^3} (-2\mathrm{i}) \frac{4\pi r^3}{3} = N \frac{\varepsilon_1}{\varepsilon}, \tag{2.27}$$

for any $r > 0$ by the Stokes theorem.

The M5-branes with different orientations either do not reduce to D4-branes or share a topological plane with the emergent D6-brane, unlike the case we just discussed. We do not understand at the moment how to incorporate the effect of these branes into the perturbative analysis of the 5d CS theory. In the present work, we will conduct the perturbative study of the 5d CS theory in the presence of the surface defect $\mathcal{S}_{N,0,0}$ only, for which the back-reaction (2.26) will be taken account of.

## 3   Algebra of local operators on defects

In the previous section, we explained that the absence of anomaly constrains the algebra of local operators on the defects. In this section, we demonstrate this explicitly by computing anomalous Feynman diagrams in the presence of a defect and deriving the OPEs of local operators as the condition that they cancel each other.

### 3.1   Associative algebra on topological line defect

The local operators on a topological line defect of the 5d $\mathfrak{gl}(1)$ Chern-Simons theory form a non-commutative associative algebra by their OPEs. The BRST-invariance of the coupling of the line defect to the 5d gauge field requires this algebra to be a representation of a certain universal associative algebra [14, 23]. For the 5d $\mathfrak{gl}(1)$ Chern-Simons theory defined on $\mathbb{R} \times \mathbb{C}^2$, the universal associative algebra is proven to be the 1-shifted affine Yangian of $\mathfrak{gl}(1)$ [23, 30], which we denote by $Y_1(\widehat{\mathfrak{gl}}(1))$.[2] See also related works in [37]. We review the definition and the relevant properties of $Y_1(\widehat{\mathfrak{gl}}(1))$ in appendix B.

Here, we will only give a brief consistency check by deriving one of the fundamental commutations relations

$$[t_{2,0}, t_{c,d}] = 2dt_{c+1,d-1} \tag{3.1}$$

of $Y_1(\widehat{\mathfrak{gl}}(1))$ as a condition for anomalous Feynman diagrams to cancel each other.

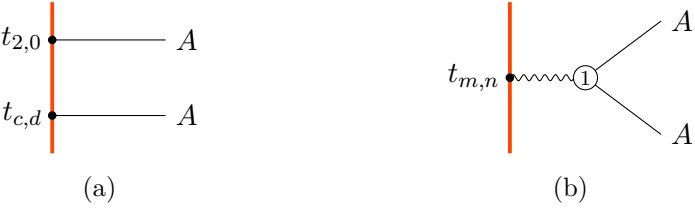

(a)                              (b)

**Figure 1**: Feynman diagrams with two external gauge fields coupling to line defect; (a) two external gauge fields directly couple to the line defect (b) two external gauge fields and a propagator from the line defect join at a bulk interaction vertex.

---

[2] The 1-shifted affine Yangian of $\mathfrak{gl}(1)$ is also known as the the deformed double current algebra. See [31–36].

Consider a generic line defect $\mathcal{L}$ located at $(0,0) \in \mathbb{C}^2$, whose coupling is established through the generalized Wilson line (2.19). Let us introduce two external gauge fields coupling to the line defect (see Figure 1a). We pick up the terms associated to $t_{2,0}$ and $t_{c,d}$ from the coupling, given by

$$\frac{\alpha_l^2}{2c!d!} \left( \int_{t_1 > t_2} t_{2,0} t_{c,d} + \int_{t_1 < t_2} t_{c,d} t_{2,0} \right) \partial_{x_1}^2 A_1 \, \partial_{x_2}^c \partial_{z_2}^d A_2, \tag{3.2}$$

where the local operators $t_{2,0}$ and $t_{c,d}$ are ordered with respect to the increasing $\mathbb{R}_t$-direction. The BRST variation of this Feynman diagram is[3]

$$\frac{\alpha_l^2}{2c!d!} \left( \int_{t_1 > t_2} t_{2,0} t_{c,d} + \int_{t_1 < t_2} t_{c,d} t_{2,0} \right) \left( \partial_{x_1}^2 d_1 c \, \partial_{x_2}^c \partial_{z_2}^d A_2 + \partial_{x_2}^c \partial_{z_2}^d d_2 c \, \partial_{x_1}^2 A_1 \right). \tag{3.3}$$

By integrating by parts, the integral only picks up the boundary terms to yield

$$\frac{\alpha_l^2}{2c!d!} \int_{\mathbb{R}_t} [t_{2,0}, t_{c,d}] \left( -\partial_x^c \partial_z^d A \, \partial_x^2 c + \partial_x^c \partial_z^d c \, \partial_x^2 A \right). \tag{3.4}$$

Next, we consider the Feynman diagram with a bulk interaction vertex, at which the two external gauge fields and a propagator from the line defect join together (see Figure 1b). The BRST variation of the diagram is evaluated to be

$$\frac{\alpha_l \varepsilon}{\varepsilon_1} \sum_{m,n=0}^{\infty} \int \frac{t_{m,n}}{m!n!} \mathrm{d}x_1 \mathrm{d}z_1 \, \partial_x^m \partial_z^n (P_{12} \, \partial_{x_1} A \, \partial_{z_1} d_1 c - P_{12} \, \partial_{z_1} A \, \partial_{x_1} d_1 c). \tag{3.5}$$

Integrating by parts and using the defining property (2.8) of the propagator, we obtain a delta-function integral which is computed to be

$$\alpha_l \varepsilon \sum_{m,n=0}^{\infty} \frac{t_{m,n}}{m!n!} \int_{\mathbb{R}_t} \partial_x^m \partial_z^n (-\partial_z A \, \partial_x c + \partial_x A \, \partial_z c). \tag{3.6}$$

Among these contributions, $m = c+1, n = d-1$ term produces the same kind of anomaly as (3.4), given by

$$\frac{\alpha_l \varepsilon}{c!(d-1)!} \int_{\mathbb{R}_t} t_{c+1,d-1} \left( -\partial_x^c \partial_z^d A \, \partial_x^2 c + \partial_x^2 A \, \partial_x^c \partial_z^d c \right). \tag{3.7}$$

Requiring the anomalous Feynman diagrams (3.4) and (3.7) to cancel each other, we precisely recover the commutation relation (3.1), provided that

$$\alpha_l = -\varepsilon. \tag{3.8}$$

---

[3]We are using $\delta_c^{\text{gauge}} A = \mathrm{d}c$, adding the holomorphic part to the variation (2.5) without affecting the integral and then dropping the commutator term. The exterior derivative is responsible for localizing the variations of the diagrams to integrals along the defect of local functions. Adding the commutator adds contributions that are either bulk integrals and/or non-local, such terms do not contribute to anomalies and must cancel among themselves.

### 3.2 Vertex algebra on holomorphic surface defect

As in the case of the operator algebra of a line defect from the previous section, the operator algebra of the holomorphic surface defect coupled to 5d CS is heavily constrained by the requirement on the coupling to be non-anomalous. In fact, given the independent complex valued parameters of the 5d CS theory, namely $\varepsilon_1$ and $\varepsilon_2$ there is a unique family of defect algebras, $\mathrm{M5}_{L,M,N}$ (2.24), labeled by non-negative integers $L, M, N$, that can be coupled to the CS theory. In the M-theory construction $L, M, N$ correspond to the numbers of M5 branes wrapping different $\Omega$-deformation planes. In this section we compute some OPE coefficients of the $\mathrm{M5}_{N,0,0}$ algebra on the defect $\mathcal{S}_{N,0,0}$. In addition to introducing the coupling (2.23), the defect M5 branes also source the background field $F^{(0)}$ (2.26) that deforms the 5d CS theory. For a choice of $N$, there is a unique background field and a unique defect algebra such that the anomalies sourced by them precisely cancel each other, rendering the coupled theory anomaly free. Given the background $F^{(0)}$ (2.26) proportional to $N$, the algebra that must be coupled to the CS theory is $\mathrm{M5}_{N,0,0} = \mathcal{W}_N$ at level $-\frac{N}{\varepsilon_1 \varepsilon_2}$.

We shall show this by checking the OPE coefficients of currents of spin up to 2 in this section, and in the next section we shall show that the defect VOA is equipped with a coproduct arising from fusion of surface defects. We start with the defect action (2.23) that couples a generic VOA to the 5d CS theory. This leads to an effective coupling between the surface defect and the bulk gauge field via quantum interaction. Concretely, this effective coupling is generated by Feynman diagrams of the form:

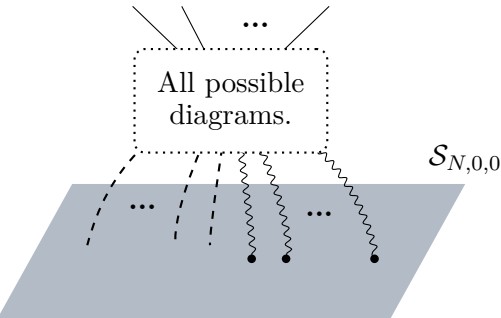

**Figure 2**: A generic diagram coupling the chiral defect $\mathcal{S}_{N,0,0}$ to the 5d CS theory.

On the external edges we have the bulk gauge field $A$, or holomorphic derivatives of it. At the end of the propagators on the defect (with the dots) we have defect local operators. A dashed line means a copy of the background connection $A^{(0)}$ with field strength given by (2.26). Such a *connected* diagram with $n$ external edges contribute a term to the effective action for the guage field that couples $n$ copies of the gauge field (or various holomorphic derivatives thereof) to the product of the defect local operators. Requiring that the sum of all such connected diagrams be gauge invariant puts strong constraints on the OPE of the defect local operators. Since a VOA has only a binary operator product and no non-trivial higher product, it will suffice to consider diagrams with only two external edges. In the rest

of this section we impose gauge invariance on the relevant diagrams to constrain the OPE of spin 1 and 2 currents.

### 3.2.1 $JJ$ OPE

Let $J := -W^{(1)}$ be the spin-1 current in the defect VOA and $\psi_0$ the level, so that the currents have the OPE:

$$J(w)J(z) \sim \frac{\psi_0}{(z-w)^2}. \tag{3.9}$$

A non-zero level makes the classical coupling anomalous due to the non-zero gauge variation of the following diagram:

$$G_{JJ}^{(0)} := \quad\qquad = \frac{\alpha_s^2}{2} \int_{\mathbb{C}_w \times \mathbb{C}_z} \mathrm{d}^2 w \mathrm{d}^2 z J(w)J(z)A_{\bar{w}}(w,\bar{w})A_{\bar{z}}(z)(z,\bar{z}). \tag{3.10}$$

The OPE (3.9) implies that this diagram varies as:

$$
\begin{aligned}
\delta_c^{\mathrm{gauge}} G_{JJ}^{(0)} &= \alpha_s^2 \int_{\mathbb{C}_w \times \mathbb{C}_z} \mathrm{d}^2 w \mathrm{d}^2 z J(w)J(z)\delta_c^{\mathrm{gauge}} A_{\bar{w}}(w,\bar{w})A_{\bar{z}}(z)(z,\bar{z}) \\
&= \alpha_s^2 \int_{\mathbb{C}_w \times \mathbb{C}_z} \mathrm{d}^2 w \mathrm{d}^2 z \frac{\psi_0}{(z-w)^2} \partial_{\bar{w}} c(w,\bar{w}) A_{\bar{z}}(z)(z,\bar{z}).
\end{aligned}
\tag{3.11}
$$

Using the identity

$$\mathrm{d}^2 w \partial_{\bar{w}} \frac{1}{w} = -2\pi \mathrm{i} \delta^{(2)}(w,\bar{w}) \quad \text{where} \quad \int_{|w|^2 < r} \delta^{(2)} = 1 \quad \text{for all} \quad r > 0 \tag{3.12}$$

we get:

$$\delta_c^{\mathrm{gauge}} G_{JJ}^{(0)} = 2\pi \mathrm{i} \psi_0 \alpha_s^2 \int_{\mathbb{C}} \mathrm{d}^2 z A_{\bar{z}} \partial_z c. \tag{3.13}$$

This variation is canceled by the variation of the diagram with one bulk interaction vertex and one background field, namely:

$$G_{A_0}^{(1)} := \quad\qquad = \frac{\varepsilon}{3\varepsilon_1} \int_{\mathbb{R} \times \mathbb{C}_w \times \mathbb{C}_z} \mathrm{d}t \mathrm{d}x \mathrm{d}z \left( A^{(0)} \partial_x A \partial_z A + A \partial_x A^{(0)} \partial_z A + A \partial_x A \partial_z A^{(0)} \right). \tag{3.14}$$

Here $A^{(0)}$ is the background gauge filed whose curvature is given by (2.26). We notice that the background is independent of $z$ and $\bar{z}$ and so we can write $A^{(0)} = A_t^{(0)} \mathrm{d}t + A_{\bar{x}}^{(0)} \mathrm{d}\bar{x}$ where both $A_t^{(0)}$ and $A_{\bar{x}}^{(0)}$ are $z, \bar{z}$-independent. As a consequence

$$\mathrm{d}A^{(0)} = \left( \partial_t A_{\bar{x}}^{(0)} - \partial_{\bar{x}} A_t^{(0)} \right) \mathrm{d}t \wedge \mathrm{d}\bar{x} + \partial_x A_t^{(0)} \mathrm{d}x \wedge \mathrm{d}t + \partial_x A_{\bar{x}}^{(0)} \mathrm{d}x \wedge \mathrm{d}\bar{x}. \tag{3.15}$$

Comparing with the expression (2.26) for $F^{(0)}$ we conclude:

$$\partial_t A_{\bar{x}}^{(0)} - \partial_{\bar{x}} A_t^{(0)} = 0 \qquad \text{and} \qquad \mathrm{d}x \partial_x A^{(0)} = F^{(0)}. \tag{3.16}$$

Once we vary the diagram (3.14), we use integration by parts with respect to $\partial_x$ and use the above formula to convert the background connection to the curvature $F^{(0)}$. Further integration by parts using the exterior derivative in $\delta_c^{\text{gauge}} A = \mathrm{d}c$ and imposing $\mathrm{d}F^{(0)} = -2\pi\mathrm{i}N\frac{\varepsilon_1}{\varepsilon}$ (2.27) we find the variation:

$$\delta_c^{\text{gauge}} G_{A_0}^{(1)} = 2\pi\mathrm{i}N \int_{\mathbb{C}} \mathrm{d}^2 z A_{\bar{z}} \partial_z c. \tag{3.17}$$

This cancels the variation (3.13) once we impose:

$$\psi_0 \alpha_s^2 = -N. \tag{3.18}$$

This fixes the defect coupling constant in terms of the level. This relation reflects the freedom to rescale $J$, which rescales the OPE coefficients, while simultaneously changing the defect coupling constant. This is merely a change of basis for our VOA and does not change the VOA itself. We fix this freedom by choosing

$$\psi_0 = -\frac{N}{\varepsilon_2 \varepsilon_3}. \tag{3.19}$$

This is a standard formula for the level of $\mathcal{W}_N$ (see (A.35)). We move on to determining some other OPE coefficients and confirm that they coincide with $\mathcal{W}_N$.

### 3.2.2 $TJ$ OPE

For a generic vertex algebra, we can write the OPE between the spin 2 and the spin 1 currents as:

$$W^{(2)}(w)J(z) \sim \frac{\beta_3}{(w-z)^3} + \frac{\beta_2(z)}{(w-z)^2} + \frac{\beta_1(z)}{w-z}. \tag{3.20}$$

Here $\beta_3$ is a c-number, and $\beta_2$, $\beta_1$ are a spin 1 and a spin 2 operator respectively. These OPE coefficients are all fixed by requiring that the anomaly of the classical diagram

$$G_{TJ}^{(0)} = \quad \begin{array}{c} \rule{0pt}{0pt} \\ \hline W^{(2)} \quad J \end{array} \quad = \alpha_s^2 \int_{\mathbb{C}_w \times \mathbb{C}_z} \mathrm{d}^2 w \mathrm{d}^2 z W^{(2)}(w)J(z)\partial_x A_{\bar{w}}(w,\bar{w})A_{\bar{z}}(z,\bar{z}) \tag{3.21}$$

be canceled by the background and interactions. We can compute the variation of the above diagram similarly to that of (3.10). We replace the product $W^{(2)}(w)J(z)$ by the OPE (3.20), use integration by parts with respect to the derivative introduced by gauge transformation, and apply the formula (3.12) – we end up with:

$$\delta_c^{\text{gauge}} G_{TJ}^{(0)} = 2\pi\mathrm{i}\alpha_s^2 \int \mathrm{d}^2 z \bigg( \frac{\beta_3}{2}(\partial_z^2 A_{\bar{z}}\partial_x c - \partial_z^2 c\partial_x A_{\bar{z}}) + \beta_2(A_{\bar{z}}\partial_z\partial_x c - c\partial_z\partial_x A_{\bar{z}}) + \beta_1(A_{\bar{z}}\partial_x c - c\partial_x A_{\bar{z}}) \bigg). \tag{3.22}$$

Only the diagram with one order-$\varepsilon$ interaction vertex (2.18a) in the bulk and a propagator sourcing $J$ on the defect can produce the same anomaly terms. The diagram is:

$$G_J^{(1)} =$$

$$= \alpha_s \frac{\varepsilon}{3\varepsilon_1} \int_{\mathbb{C}_w \times M_v} \mathrm{d}w \wedge \mathrm{d}x \wedge \mathrm{d}z J(w) \wedge \left( \overbrace{A_{\bar{w}}(w,\bar{w})\mathrm{d}\bar{w} \wedge A}(v) \wedge \partial_x A(v) \wedge \partial_z A(v) \right.$$

$$+ \overbrace{A_{\bar{w}}(w,\bar{w})\mathrm{d}\bar{w} \wedge A(v) \wedge \partial_x A}(v) \wedge \partial_z A(v)$$

$$\left. + \overbrace{A_{\bar{w}}(w,\bar{w})\mathrm{d}\bar{w} \wedge A(v) \wedge \partial_x A(v) \wedge \partial_z A}(v) \right). \tag{3.23}$$

Here $M_v = \mathbb{R} \times \mathbb{C}_x \times \mathbb{C}_z$ is the 5d space-time with the coordinate $v = (t, x, \bar{x}, z, \bar{z})$. The Wick contraction produces the 2-point correlation function or equivalently the propagator $P$ via (2.14). To compute its gauge variation we shall have to use the defining property (2.8) of the propagator. Let us look at the variation of the first term above in details.

$$- \delta_c^{\mathrm{gauge}} \left( \alpha_s \frac{\varepsilon}{3\varepsilon_1} \int_{\mathbb{C}_w \times M_v} \mathrm{d}w \wedge \mathrm{d}x \wedge \mathrm{d}z J(w) \wedge P(w,v) \wedge \partial_x A(v) \wedge \partial_z A(v) \right)$$

$$= -\alpha_s \frac{\varepsilon}{3\varepsilon_1} \int_{\mathbb{C}_w \times M_v} \mathrm{d}w \wedge \mathrm{d}x \wedge \mathrm{d}z J(w) \wedge P(w,v) \left( \partial_x \mathrm{d}c(v) \wedge \partial_z A(v) + \partial_x A(v) \wedge \partial_z \mathrm{d}c(v) \right)$$

$$= \alpha_s \frac{\varepsilon}{3} \int_{\mathbb{C}_w \times M_v} \mathrm{d}w J(w) \wedge \delta^{(5)}(w-v) \wedge \left( \partial_x c(v) \partial_z A(v) - \partial_x A(v) \partial_z c(v) \right)$$

$$= -\frac{\alpha_s \varepsilon}{3} \int_{\mathbb{C}} \mathrm{d}^2 z J(A_{\bar{z}}\partial_z\partial_x c - c\partial_z\partial_x A_{\bar{z}}) - \frac{\alpha_s \varepsilon}{3} \int_{\mathbb{C}} \mathrm{d}^2 z \partial_z J(A_{\bar{z}}\partial_x c - c\partial_x A_{\bar{z}}). \tag{3.24}$$

In the second equality above we have done integration by parts with respect to the exterior derivative and applied (2.8). In doing this integration by parts we have imposed the equation of motion $\mathrm{d}A(v) = 0$ on the fields attached to the external legs of the diagram and we have further used that $\mathrm{d}w \wedge \mathrm{d}J(w) = 0$ by holomorphicity. In the last equality we have used the delta 5-form to reduce the variation to an integration over the defect only and used more integrations by parts to rewrite the expression in a way as to make comparison with (3.22) easier.

We can similarly compute the gauge variations of the remaining two terms in (3.23), combining all of them we find that the variation of $G_J^{(1)}$ is three times the variation of the first term, i.e.:

$$\delta_c^{\mathrm{gauge}} G_J^{(1)} = -\alpha_s \varepsilon \int_{\mathbb{C}} \mathrm{d}^2 z J(A_{\bar{z}}\partial_z\partial_x c - c\partial_z\partial_x A_{\bar{z}}) - \alpha_s \varepsilon \int_{\mathbb{C}} \mathrm{d}^2 z \partial_z J(A_{\bar{z}}\partial_x c - c\partial_x A_{\bar{z}}). \tag{3.25}$$

We readily recognize the coefficients of $\beta_2$ and $\beta_1$ from (3.22) in the above equation, whereas the coefficient of $\beta_3$ is missing. There are in fact no diagrams whose variations can give rise

to the $\beta_3$-term from (3.22). Therefore, we have $\delta_c^{\text{gauge}}\left(G_{TJ}^{(0)} + G_J^{(1)}\right) = 0$ given:

$$\beta_3 = 0, \qquad \beta_2 = J, \qquad \beta_1 = \partial_z J, \qquad \varepsilon = 2\pi i \alpha_s = 2\pi i \sqrt{-\frac{N}{\psi_0}}. \qquad (3.26)$$

In the last equality above we have used the previously derived constraint (3.18). The expressions for the $\beta_1, \beta_2$, and $\beta_3$ implies that the OPE (3.20) has precisely the form of the OPE between the stress energy tensor and a spin-1 current in the $\mathcal{W}_N$-algebra (A.39).

### 3.2.3  $TT$ OPE

Our next task, naturally, is to see whether the $W^{(2)}W^{(2)}$ OPE is indeed constrained to coincide with the $TT$ OPE of $\mathcal{W}_N$. The OPE of some generic spin-2 currents can be written as:

$$W^{(2)}(w)W^{(2)}(z) \sim \frac{\gamma_4}{(w-z)^4} + \frac{\gamma_3(z)}{(w-z)^3} + \frac{\gamma_2(z)}{(w-z)^2} + \frac{\gamma_1(z)}{w-z}. \qquad (3.27)$$

This means that the classical diagram

$$G_{TT}^{(0)} = \quad \begin{array}{c} \includegraphics \\ W^{(2)} \quad W^{(2)} \end{array} \quad = \frac{\alpha_s^2}{2}\int_{\mathbb{C}_w \times \mathbb{C}_z} \mathrm{d}^2 w \mathrm{d}^2 z\, W^{(2)}(w)W^{(2)}(z)\partial_x A_{\bar{w}}\partial_x A_{\bar{z}} \qquad (3.28)$$

has the gauge variation

$$\delta_c^{\text{gauge}} G_{TT}^{(0)} = 2\pi i \alpha_s^2 \int_{\mathbb{C}} \mathrm{d}^2 z \left(\frac{\gamma_4}{6}\partial_z^3 \partial_x c + \frac{\gamma_3}{2}\partial_z^2 \partial_x c + \gamma_2 \partial_z \partial_x c + \gamma_1 \partial_x c\right)\partial_x A_{\bar{z}}. \qquad (3.29)$$

There are no candidate diagrams whose anomalies can cancel the $\gamma_3$-term above and therefore we must have:

$$\gamma_3 = 0. \qquad (3.30)$$

There are possible diagrams whose anomalies can cancel the $\gamma_4$ terms and computing them will fix the central charge of the VOA. This is not necessary for identifying the VOA and we leave it for future work, in the present work once we identify the algebra as $\mathcal{W}_N$ we can also fix the central charge by its level. For now we work out the diagrams that contribute to canceling the $\gamma_1$ and $\gamma_2$ terms. The relevant diagram has an order-$\varepsilon$ bulk interaction vertex and it sources $W^{(2)}$ on the defect, namely:

$$G_T^{(1)} = \quad \begin{array}{c} \includegraphics \\ W^{(2)} \end{array}$$

$$= \frac{\alpha_s \varepsilon}{3\varepsilon_1}\int_{\mathbb{C}_w \times M_v} \mathrm{d}w \wedge \mathrm{d}x \wedge \mathrm{d}z\, W^{(2)}(w) \wedge \left(\partial_{x_w}\overline{A_{\bar{z}}(w,\bar{w})\mathrm{d}\bar{z} \wedge A}(v) \wedge \partial_x A(v) \wedge \partial_z A(v)\right.$$

$$+ \partial_{x_w}\overline{A_{\bar{z}}(w,\bar{w})\mathrm{d}\bar{z} \wedge A(v) \wedge \partial_x A}(v) \wedge \partial_z A(v)$$

$$\left. + \partial_{x_w}\overline{A_{\bar{z}}(w,\bar{w})\mathrm{d}\bar{z} \wedge A(v) \wedge \partial_x A(v) \wedge \partial_z A}(v)\right). \qquad (3.31)$$

Here by $\partial_{x_w}$ we mean derivative with respect to the $x$-coordinate at the end of the propagator attached to the defect, as opposed to the $x$-coordinate of the end attached to the bulk interaction vertex. As in (3.23), $M_v = \mathbb{R} \times \mathbb{C}_x \times \mathbb{C}_z$ and $v = (t, x, \bar{x}, z, \bar{z})$. One can check that the gauge variation of the full diagram is simply three times the variation of the first line above, so that we get:

$$
\begin{aligned}
\delta_c^{\text{gauge}} G_T^{(1)} &= 3\delta_c^{\text{gauge}} \left( \frac{\alpha_s \varepsilon}{3\varepsilon_1} \int_{\mathbb{C}_w \times M_v} \mathrm{d}w \wedge \mathrm{d}x \wedge \mathrm{d}z W^{(2)}(w) \wedge \partial_x P(w, v) \wedge \partial_x A(v) \wedge \partial_z A(v) \right) \\
&= \alpha_s \varepsilon \int_{\mathbb{C}} \mathrm{d}^2 z W^{(2)}(z) (\partial_x^2 c \partial_z A_{\bar{z}} - \partial_x^2 A_{\bar{z}} \partial_z c) - 2\alpha_s \varepsilon \int_{\mathbb{C}} \mathrm{d}^2 z W^{(2)}(z) \partial_z \partial_x c \partial_x A_{\bar{z}} \\
&\quad - \alpha_s \varepsilon \int_{\mathbb{C}} \mathrm{d}^2 z \partial_z W^{(2)}(z) \partial_x c \partial_x A_{\bar{z}}.
\end{aligned}
\tag{3.32}
$$

In writing the first line above we have used the relation (2.14) between 2-point correlation function and the propagator and also changed the $x$-derivative from acting on the defect end of the propagator to the bulk end. The two signs introduced by these two operations cancel each other. The rest of the computation is similar to (3.24). We observe that when the variations (3.29) and (3.32) are added up, the last two terms of (3.32) cancel the last two terms of (3.29) given:

$$
\gamma_2 = 2W^{(2)}, \qquad \gamma_1 = \partial_z W^{(2)}, \qquad \varepsilon = 2\pi \mathrm{i} \alpha_s.
\tag{3.33}
$$

Note that the relation between $\varepsilon$ and $\alpha_s$ is the same as the one coming from the anomaly cancellation of the TJ OPE (3.26), of course, a different relation would imply the nonexistence of a consistent coupling between the 5d CS theory and the surface defect. Additionally, the values of $\gamma_1, \gamma_2$, and $\gamma_3$ (from (3.30)) imply that the $W^{(2)} W^{(2)}$ OPE (3.27) of the defect VOA is the same as the $TT$ OPE of a $\mathcal{W}$-algebra where $T$ is the stress energy tensor (A.39). The central charge $\gamma_4$ of the $\mathcal{W}$-algebra must be fixed by computing further diagrams that can cancel the $\gamma_4$ term from (3.29) which we have not done in this work. We also leave for future work evaluations of diagrams canceling the first term of (3.32) which will involve spin-3 currents. For our present purposes we only need to know that the spin-1 and spin-2 currents of our defect VOA can be identified with the corresponding currents of a $\mathcal{W}_N$.

## 4 Coproduct and defect fusion

The same kind of defects − line or surface − may lie on top of each other and fuse into a single defect. The coupling of the fused defect is built from the couplings of the individual defects before the fusion, equipping the algebra of local operators on the defect with a coproduct. We will provide a first-principle derivation of the coproduct by a direct perturbative study of the defect fusion.[4]

---

[4] We find that the analysis in [38] contains errors.

### 4.1 Fusion of line defects

Let us introduce two line defects $\mathcal{L}$ and $\mathcal{L}'$, whose locations on the transverse holomorphic planes $\mathbb{C}_x \times \mathbb{C}_z$ are $(0,0)$ and $(0,z)$, respectively. The two line defects can approach each other and fuse into a single line defect $\mathcal{L} \circ_z \mathcal{L}'$. The coupling of the fused line defect can be computed by evaluating the Feynman diagrams with a single external gauge field before the fusion, which are collected into the form of

$$\sum_{m,n=0}^{\infty} \frac{\alpha_l}{m!n!} \int_{\mathbb{R}_t} \mathrm{d}t \, \Delta(z)(t_{m,n}) \partial_x^m \partial_z^n A_t. \tag{4.1}$$

Here, $\Delta(z)(t_{m,n})$ is the local operator on the fused line defect built from the local operators on the two line defects before the fusion, coupling to a fixed ghost mode. Therefore, they form a representation of the 1-shifted affine Yangian $Y_1(\widehat{\mathfrak{gl}}(1))$ by the general principle of the Koszul duality [20, 23]. In other words, the fusion of the line defects gives rise to a meromorphic coproduct $\Delta(z) : Y_1(\widehat{\mathfrak{gl}}(1)) \to Y_1(\widehat{\mathfrak{gl}}(1)) \,\widehat{\otimes}\, Y_1(\widehat{\mathfrak{gl}}(1))(\!(z)\!)$, and we have

$$\rho_{\mathcal{L} \circ_z \mathcal{L}'} = (\rho_{\mathcal{L}} \otimes \rho_{\mathcal{L}'})\Delta(z). \tag{4.2}$$

As reviewed in appendix B, the 1-shifted affine Yangian of $\mathfrak{gl}(1)$ is indeed equipped with a meromorphic coproduct which is fully determined by

$$\Delta(z)(t_{0,m}) = t_{0,m} \otimes 1 + 1 \otimes \sum_{n=0}^{m} \binom{m}{n} z^{m-n} t_{0,n}, \tag{4.3a}$$

$$\Delta(z)(t_{2,0}) = t_{2,0} \otimes 1 + 1 \otimes t_{2,0} + 2\sigma_3 \sum_{m,n=0}^{\infty} \frac{(m+n+1)!}{m!n!}(-1)^n z^{-m-n-2} t_{0,m} \otimes t_{0,n}. \tag{4.3b}$$

Here, we will confirm that this meromorphic coproduct indeed governs the fusion of two line defects by directly evaluating the Feynman diagrams (4.1) relevant to the two elements in (4.3).

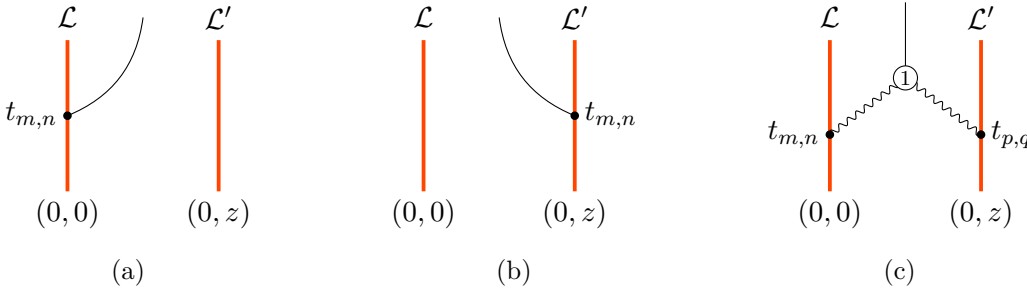

**Figure 3**: Feynman diagrams for fusion of line defects. The two line defects $\mathcal{L}$ and $\mathcal{L}'$ are located at $(0,0)$ and $(0,z)$ on the holomorphic planes $\mathbb{C}_x \times \mathbb{C}_z$, respectively.

The first relation (4.3a) is straightforward. There are only two tree diagrams in which the external gauge field couples to one of the two line defects $\mathcal{L}$ and $\mathcal{L}'$ (see Figure 3a and

[3b](). There cannot be any bulk interaction vertex since such diagrams produce $m \geq 1$ terms in [(4.1)](). The first diagram is evaluated to be

$$\sum_{m=0}^{\infty} \frac{\alpha_l}{m!} \int_{\mathbb{R}_t} \mathrm{d}t \, (t_{0,m} \otimes 1) \partial_z^m A_t. \tag{4.4}$$

The second diagram provides a similar contribution, only with the external gauge field expanded around $z = 0$:

$$
\begin{aligned}
\sum_{n=0}^{\infty} \frac{\alpha_l}{n!} \int_{\mathbb{R}_t} \mathrm{d}t \, (1 \otimes t_{0,n}) \partial_z^n \sum_{m=0}^{\infty} \frac{z^m}{m!} \partial_z^m A_t(0) &= \sum_{n \leq m} \frac{\alpha_l}{n!(m-n)!} \int_{\mathbb{R}_t} \mathrm{d}t (1 \otimes t_{0,n}) z^{m-n} \partial_z^m A_t \\
&= \sum_{m=0}^{\infty} \frac{\alpha_l}{m!} \int_{\mathbb{R}_t} \mathrm{d}t \sum_{n=0}^{m} \frac{m!}{n!(m-n)!} (1 \otimes t_{0,n}) z^{m-n} \partial_z^m A_t.
\end{aligned}
\tag{4.5}
$$

Summing the two contributions, the relevant term in the coupling of the fused line defect $\mathcal{L} \circ_z \mathcal{L}'$ reads

$$\sum_{m=0}^{\infty} \frac{\alpha_l}{m!} \int_{\mathbb{R}_t} \mathrm{d}t \, \Delta(z)(t_{0,m}) \partial_z^m A_t, \tag{4.6}$$

where $\Delta(z)(t_{0,m})$ precisely recovers the image [(4.3a)]() of the generator $t_{0,m}$ under the meromorphic coproduct.

The second relation [(4.3b)]() is more involved. There are still two tree diagrams similar to the previous case, which provide the obvious contribution

$$\frac{\alpha_l}{2} \int_{\mathbb{R}_t} \mathrm{d}t (t_{2,0} \otimes 1 + 1 \otimes t_{2,0}) \partial_x^2 A_t. \tag{4.7}$$

In addition, there is a one-loop diagram with a bulk interaction vertex (see Figure [3c]()). For the interaction vertex, only the first-order term in the expansion of the Moyal product contributes since higher-order terms provide $m \geq 3$ terms in [(4.1)](). This diagram is evaluated to be

$$\alpha_l^2 \frac{\varepsilon}{\varepsilon_1} \sum_{m,n,p,q=0}^{\infty} \frac{t_{m,n} \otimes t_{p,q}}{m!n!p!q!} \int \mathrm{d}x_0 \mathrm{d}z_0 \, \partial_{x_1}^m \partial_{z_1}^n \partial_{x_2}^p \partial_{z_2}^q \left( -A_0 \partial_{x_0} P_{10} \partial_{z_0} P_{20} + A_0 \partial_{x_0} P_{20} \partial_{z_0} P_{10} \right), \tag{4.8}$$

where all the products are the wedge products of differential forms. The two terms in the parenthesis can be computed separately. For the first term, let us compute the relevant holomorphic derivatives of the propagators as

$$
\begin{aligned}
\partial_{x_1}^m \partial_{z_1}^n \partial_{x_0} P_{10} &= -\frac{3\varepsilon_1}{16\pi^2} \frac{\Gamma\left(\frac{7}{2} + m + n\right)}{\Gamma\left(\frac{5}{2}\right)} \frac{\bar{x}_0^{m+1} \bar{z}_0^n \mathrm{d}t_1 (-\bar{x}_0 \mathrm{d}\bar{z}_0 + \bar{z}_0 \mathrm{d}\bar{x}_0)}{(t_1^2 + |x_0|^2 + |z_0|^2)^{\frac{7}{2} + m + n}} \\
\partial_{x_2}^p \partial_{z_2}^q \partial_{z_0} P_{20} &= -\frac{3\varepsilon_1}{16\pi^2} \frac{\Gamma\left(\frac{7}{2} + m + n\right)}{\Gamma\left(\frac{5}{2}\right)} (-1)^q \frac{\bar{x}_0^p \bar{z}_{20}^{q+1} \mathrm{d}t_2 (\bar{x}_0 \mathrm{d}\bar{z}_0 + \bar{z}_{20} \mathrm{d}\bar{x}_0)}{(t_1^2 + |x_0|^2 + |z_{20}|^2)^{\frac{7}{2} + p + q}}.
\end{aligned}
\tag{4.9}
$$

Note that their product is proportional to $\bar{x}_0^{m+p+2}$. The two-dimensional $x_0$-plane integral vanishes unless the external gauge field provides $\frac{x_0^{m+p+2}}{(m+p+2)!}\partial_{x_0}^{m+p+2}A_0$ from its Taylor expansion around $x_0 = 0$, cancelling the angular dependence of the integrand. However, we are sorting out the second-derivative $\partial_{x_0}^2 A_0$ here, to keep the contributions to $\Delta(z)(t_{2,0})$ in (4.1) only. Thus, we notice that the only contribution comes from $m = p = 0$.

Then, the integral reads (after changing the dummy indices)

$$
-\alpha_l^2 \varepsilon_1 \varepsilon \sum_{m,n=0}^{\infty} \frac{t_{0,m} \otimes t_{0,n}}{m!n!} \left(\frac{3}{16\pi^2}\right)^2 \frac{\Gamma\left(\frac{7}{2}+m\right)\Gamma\left(\frac{7}{2}+n\right)}{\Gamma\left(\frac{5}{2}\right)^2} (-1)^n \times \frac{1}{2}
$$
$$
\times \int d^2 x_0 d^2 z_0\, \partial_{x_0}^2 A_0 \frac{|x_0|^4 \bar{z}_2 \bar{z}_0^m \bar{z}_{20}^{n+1} dt_1 dt_2}{(t_1^2 + |x_0|^2 + |z_0|^2)^{\frac{7}{2}+m}(t_2^2 + |x_0|^2 + |z_{20}|^2)^{\frac{7}{2}+n}}.
$$
(4.10)

It is straightforward to perform the $t_1$- and $t_2$-integrals. Then, we introduce the Feynman parametrization,

$$
\frac{1}{A^\alpha B^\beta} = \frac{\Gamma(\alpha+\beta)}{\Gamma(\alpha)\Gamma(\beta)} \int_0^1 dy \frac{y^{\alpha-1}(1-y)^{\beta-1}}{(yA + (1-y)B)^{\alpha+\beta}},
$$
(4.11)

to express the remaining integral as

$$
-\alpha_l^2 \varepsilon_1 \varepsilon \sum_{m,n=0}^{\infty} \frac{t_{0,m} \otimes t_{0,n}}{m!n!} \frac{1}{32\pi^4}(-1)^n(m+n+5)!
$$
$$
\times \int d^2 x_0 d^2 z_0\, \partial_{x_0}^2 A_0\, dy \frac{(1-y)^{m+2}y^{n+2}|x_0|^4 \bar{z}_2 \bar{z}_0^m \bar{z}_{20}^{n+1}}{(|z_0 - y z_2|^2 + |x_0|^2 + y(1-y)|z_2|^2)^{m+n+6}}.
$$
(4.12)

Now, we change the integration variables by $z_0 \to z_0 + y z_2$. In the numerator of the integrand, we have $(\bar{z}_0 + y \bar{z}_2)^m((1-y)\bar{z}_2 - \bar{z}_0)^{n+1}$ as a consequence. The only non-zero contribution comes from $y^m(1-y)^{n+1}\bar{z}_2^{m+n+1}$ in its expansion, since the angular part of the $z_0$-plane integral vanishes for other terms. With this, we perform the $x_0$- and $z_0$-integrals to get

$$
-\alpha_l^2 \varepsilon_1 \varepsilon \sum_{m,n=0}^{\infty} \frac{(m+n+5)!}{m!n!} \frac{(-1)^n}{32\pi^4} \times (-2\mathrm{i})^2 \times (2\pi)^2
$$
$$
\times \frac{1}{2}\frac{(m+n+1)!}{(m+n+5)!} z_2^{-m-n-2} \int_{\mathbb{R}_t} (t_{0,m} \otimes t_{0,n})\partial_{x_0}^2 A_0 \int_0^1 dy\,(1-y).
$$
(4.13)

The $y$-integral simply gives $\frac{1}{2}$, and we finally arrive at (after omitting unnecessary subscripts)

$$
\frac{\alpha_l^2 \varepsilon_1 \varepsilon}{8\pi^2} \sum_{m,n=0}^{\infty} \frac{(m+n+1)!}{m!n!} \int_{\mathbb{R}_t} dt\,(t_{0,m} \otimes t_{0,n})(-1)^n z^{-m-n-2}\partial_x^2 A_t.
$$
(4.14)

The second term in the parenthesis of (4.8) can be evaluated in a similar way, and it turns out to give exactly the same contribution as (4.14). All in all, we arrive at

$$
\frac{\alpha_l}{2} \int_{\mathbb{R}_t} dt\,\Delta(z)(t_{2,0})\partial_x^2 A_t,
$$
(4.15)

where $\Delta(z)(t_{2,0})$ precisely recovers the expression (4.3b) from the meromorphic coproduct. Importantly, the structure constant $\sigma_3 = \varepsilon_1\varepsilon_2\varepsilon_3$ is exactly reproduced since

$$\sigma_3 = \frac{\alpha_l\varepsilon_1\varepsilon}{4\pi^2}, \tag{4.16}$$

by $\varepsilon = 2\pi\sqrt{-\varepsilon_2\varepsilon_3}$ (3.26) and $\alpha_l = -2\pi\sqrt{-\varepsilon_2\varepsilon_3}$ (3.8) determined from the anomaly cancellation.

## 4.2 Fusion of surface defects

We now consider the case of two surface defects, $\mathcal{S}_{N_1,0,0}$ and $\mathcal{S}_{N_2,0,0}$. Let us order these defects in the $\mathbb{R}_t$-direction by placing the first at $t_1 = 0$ and the second at $t_2 = \epsilon > 0$. Next, we bring the two surface defects on top of each other by taking the limit $\epsilon \to 0$. As a result, the two surface defects fuse into a new surface defect, which we denote by $\mathcal{S}_{N_1+N_2,0,0} = \mathcal{S}_{N_1,0,0} \circ \mathcal{S}_{N_2,0,0}$. Summing over all the Feynman diagrams with an external gauge field into the form of

$$\sum_{m=0}^{\infty} \frac{\alpha_s}{m!} \int_{\mathbb{C}_z} \mathrm{d}^2z\, \Delta(W^{(m+1)}(z))\partial_x^m A_{\bar{z}}, \tag{4.17}$$

before the fusion, we can determine the coupling of the emergent surface defect after the fusion.

Let us recall that the $\mathcal{W}_\infty$-algebra is endowed with a coproduct $\Delta : \mathcal{W}_\infty \to \mathcal{W}_\infty \otimes \mathcal{W}_\infty$, which is a vertex algebra homomorphism (see appendix A.5). We will demonstrate that the surface defect fusion is governed by this coproduct, namely, the local operators appearing in the coupling (4.17) of the fused surface defect is its image under the coproduct mapped through the representations $\rho_{\mathrm{M5}_{N_1,0,0}} \otimes \rho_{\mathrm{M5}_{N_2,0,0}}$. In other words, we have

$$\rho_{\mathcal{S}\circ\mathcal{S}'} = (\rho_\mathcal{S} \otimes \rho_{\mathcal{S}'})\Delta, \tag{4.18}$$

in general. Here, we specialize to $\mathcal{S} = \mathcal{S}_{N_1,0,0}$ and $\mathcal{S}' = \mathcal{S}_{N_2,0,0}$.

It is crucial to correctly identify which combinations of currents in $\mathcal{W}_\infty$ appear in the coupling, and we explicitly confirmed that they are the elements in the primary basis in the previous section. The image of the first two currents under the coproduct composed with the representations $\rho_{\mathrm{M5}_{N_1,0,0}} \otimes \rho_{\mathrm{M5}_{N_2,0,0}}$ is given by

$$\Delta(J(z)) = J(z) \otimes 1 + 1 \otimes J(z) \tag{4.19a}$$

$$\Delta(T(z)) = T(z) \otimes 1 + 1 \otimes T(z) + \frac{\varepsilon_1}{2}\left(N_2(\partial J(z) \otimes 1) - N_1(1 \otimes \partial J(z))\right). \tag{4.19b}$$

We will confirm that this coproduct indeed describes the fusion of the two surface defects by evaluating the relevant Feynman diagrams.

The confirmation of the first relation (4.19a) is trivial. There are only two tree diagrams in which the external gauge field couples to one of the two surface defects (see Figure 4). There cannot be any bulk interaction vertex since the diagram would then produce $m \geq 1$

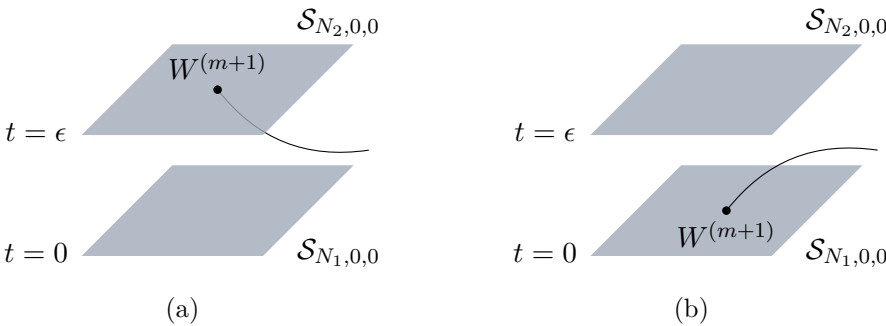

(a)             (b)

**Figure 4**: Tree diagrams for fusion of surface defects

terms in (4.17).

We turn to the second relation (4.19b). There are two tree diagrams as in the previous case which provide the obvious contribution (see Figure 4)

$$\alpha_s \int_{\mathbb{C}_z} \mathrm{d}^2 z \ (T(z) \otimes 1 + 1 \otimes T(z)) \, \partial_x A_{\bar{z}}. \tag{4.20}$$

In addition, there are one-loop diagrams having one bulk interaction vertex. The one-loop diagram with two propagators coming from the two defects joining at the interaction vertex turns out to produce the terms in (4.17) with $m \geq 2$, so we do not need to consider them for our computation of $\Delta(T(z))$ for which $m = 1$. Instead, there are two diagrams in which the back-reaction field from one of the defects and a propagator from the other defect joins at the interaction vertex, together with the external gauge field (see Figure 5). Only the first term in the Moyal product expansion contributes, since higher-order terms yield $m \geq 2$ terms in (4.17).

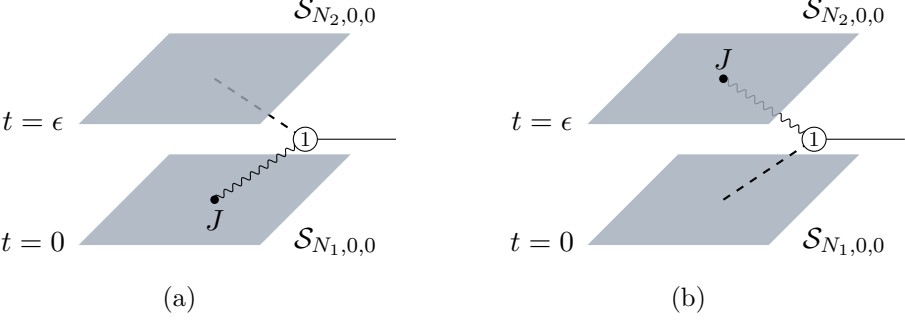

(a)             (b)

**Figure 5**: One-loop diagrams contributing to $\Delta(T(z))$. The two surface defects $\mathcal{S}_{N_1,0,0}$ and $\mathcal{S}_{N_2,0,0}$ are located at $t = 0$ and $t = \epsilon > 0$ on $\mathbb{R}_t$, respectively. The limit $\epsilon \to 0$ is taken to initiate the fusion.

It is straightforward to evaluate the first diagram as

$$-\frac{\alpha_s \varepsilon}{\varepsilon_1} \int dz_1 (J(z_1) \otimes 1) dz_0 \, P_{10} \, F_{02}^{(0)} \partial_{z_0} A_0, \tag{4.21}$$

where $F_{02}^{(0)}$ indicates the back-reaction from the second stack of $N_2$ D4-branes at the interaction vertex. We change the integration variable by $z_1 \rightarrow z_1 + z_0$. Note that $J(z_1) = \sum_n J_{-n-1} z_1^n$ is modified to $\sum_n J_{-n-1}(z_1 + z_0)^n$. The only non-zero contribution comes from $J(z_0) = \sum_n J_{-n-1} z_0^n$ in its expansion, since the angular part of the $z_1$-integral vanishes for other terms. Thus, the $z_1$-integral can be performed to yield

$$\frac{1}{2\pi i} \alpha_s \varepsilon \int dz_0 \, (J(z_0) \otimes 1) P'_{10} \, F_{02}^{(0)} \partial_{z_0} A_0, \tag{4.22}$$

where we defined the integrated propagator as

$$P'(t, x, \bar{x}) = \frac{1}{2} \frac{-\frac{1}{2} t d\bar{x} + \bar{x} dt}{(t^2 + |x|^2)^{\frac{3}{2}}} = -\frac{1}{4} \left( -d\bar{x}\partial_t + 4 dt \partial_x \right) (t^2 + |x|^2)^{-\frac{1}{2}}. \tag{4.23}$$

We notice that the back-reaction $F_{02}^{(0)}$ can also be expressed in terms of $P'$, so that the integral is re-expressed as

$$-\frac{1}{2\pi i} N_2 \alpha_s \varepsilon_1 \int dz_0 \, (J(z_0) \otimes 1) \partial_{z_0} A \, dx_0 \, P'(t_0, x_0, \bar{x}_0) P'(t_0 - \epsilon, x_0, \bar{x}_0). \tag{4.24}$$

At this point, we use the identity[5]

$$\lim_{\epsilon \to 0} dx_0 \, P'(t_0, x_0, \bar{x}_0) \, P'(t_0 - \epsilon, x_0, \bar{x}_0) = -i\pi \partial_{x_0} \delta^{(3)}(t_0, x_0, \bar{x}_0), \tag{4.25}$$

and evaluate the three-dimensional delta-function integral to finally arrive at (after omitting unnecessary subscripts)

$$\alpha_s \frac{N_2 \varepsilon_1}{2} \int_{\mathbb{C}_z} d^2 z \, (\partial J(z) \otimes 1) \partial_x A_{\bar{z}}. \tag{4.26}$$

A similar computation shows that the opposite diagram contributes the same term with an extra sign and $N_1 \leftrightarrow N_2$ exchanged. All in all, we obtain

$$\alpha_s \int_{\mathbb{C}_z} d^2 z \, \Delta(T(z)) \partial_x A_{\bar{z}}, \tag{4.27}$$

where $\Delta(T(z))$ exactly recovers the coproduct (4.19b) for the spin-2 current in the primary basis mapped through the representation $\rho_{\text{M5}_{N_1,0,0}} \otimes \rho_{\text{M5}_{N_2,0,0}}$.

---

[5]This identity is adopted from eq. (5.5) in [24].

# 5 R-matrices and Miura operators from defect intersections

Finally, we study the configuration where a topological line defect and a holomorphic surface defect coexist. Though our actual computational setting will be the 5d CS theory on $\mathbb{R}_t \times \mathbb{C}_x \times \mathbb{C}_z$ as before, let us begin with considering the 5d CS theory on $\mathbb{R}_t \times \mathbb{C}_x \times \mathbb{C}_z^\times$ for the moment to elucidate how the intersection of the two defects gives rise to an R-matrix.

We take a generic line defect $\mathcal{L}$, supported on $\mathbb{R}_t$, and a generic surface defect $\mathcal{S}$, supported on $\mathbb{C}_z^\times$. Let us denote the algebra of local operators on $\mathcal{L}$ by M2, and the mode algebra of local operators on $\mathcal{S}$ by $U(\mathrm{M5})$. They form representations of affine Yangian of $\mathfrak{gl}(1)$, which we call $\rho_\mathcal{L} : Y(\widehat{\mathfrak{gl}}(1)) \twoheadrightarrow \mathrm{M2}$ and $\rho_\mathcal{S} : Y(\widehat{\mathfrak{gl}}(1)) \twoheadrightarrow U(\mathrm{M5})$, respectively.

The two defects intersect at a point in the 5d worldvolume, supporting a space of local operators. A local operator $R$ at the intersection should be built from the local operators on the line defect and the surface defect, and we have

$$R \in \mathrm{M2} \,\widehat{\otimes}\, U(\mathrm{M5}), \tag{5.1}$$

where the completed tensor product is used to allow linear combination of infinitely many terms. Equivalently, the local operator can be viewed as acting on the space $\mathcal{V} \otimes \mathcal{F}$, where $\mathcal{V}$ is the Hilbert space of the worldline theory on the line defect attached to the negative infinity of $\mathbb{R}_t$ and $\mathcal{F}$ is the Hilbert space of the worldvolume theory on the surface defect attached to the origin of $\mathbb{C}_z^\times$. It produces another state in the same space, attached to the positive infinity of $\mathbb{R}_t$ and the infinity of $\mathbb{C}_z^\times$. Thus, we have $R \in \mathrm{End}(\mathcal{V}) \,\widehat{\otimes}\, \mathrm{End}(\mathcal{F}) = \mathrm{M2} \,\widehat{\otimes}\, U(\mathrm{M5})$.

We can slightly modify the configuration by translating the line defect to $\delta \in \mathbb{C}_x$ along the holomorphic plane transverse to the surface defect (located at the origin $0 \in \mathbb{C}_x$). The algebra of local operators on the line defect should remain the same, but how they couple to the 5d CS theory alters since the location of the line defect alters the mode expansion of the gauge field that the local operators pair with. Let us denote the local operators on the translated line defect by $t'_{m,n}$ from which the coupling is written as

$$\mathrm{Pexp} \sum_{m=0}^\infty \sum_{n\in\mathbb{Z}} \alpha_l \int_{\mathbb{R}_t} \frac{t'_{m,n}}{m!} \partial_x^m (A_t)_n, \tag{5.2}$$

where $(A_t)_n$ is the mode of the gauge field under the Laurent expansion on $\mathbb{C}_z^\times$. Note that the gauge field $A_t$ is evaluated at the location of the line defect, $\delta \in \mathbb{C}_x$. Now, we can re-expand the gauge field around $0 \in \mathbb{C}_x$ to get

$$\mathrm{Pexp} \sum_{m=0}^\infty \sum_{n\in\mathbb{Z}} \alpha_l \int_{\mathbb{R}_t} \frac{1}{m!} \left( \sum_{l=0}^m \frac{m!}{l!(m-l)!} \delta^{m-l} t'_{l,n} \right) \partial_x^m (A_t)_n. \tag{5.3}$$

This implies that the translation of the line defect induces an automorphism of $Y(\widehat{\mathfrak{gl}}(1))$ given by

$$t_{m,n} \mapsto \sum_{l=0}^m \frac{m!}{l!(m-l)!} \delta^{m-l} t_{l,n}. \tag{5.4}$$

It is easy to see that this automorphism is in fact an inner automorphism of conjugating by $e^{\delta t_{0,1}}$. Indeed, a straightforward computation shows that

$$t_{m,n} \mapsto e^{-\delta t_{0,1}} t_{m,n} e^{\delta t_{0,1}} = \sum_{l=0}^{\infty} \frac{(-\delta)^l \mathrm{ad}_{t_{0,1}}^l (t_{m,n})}{l!} = \sum_{l=0}^{m} \frac{m!}{l!(m-l)!} \delta^{m-l} t_{l,n}, \qquad (5.5)$$

where we used the commutation relation $[t_{0,1}, t_{m.n}] = -m t_{m-1,n}$ (see appendix B).[6]

Thus, the effect of translating the line defect to $\delta \in \mathbb{C}_x$ on the local operator $R$ at the intersection is to conjugate it by $e^{\delta(t_{0,1} \otimes \mathrm{id})}$. The so-obtained $R(\delta)$ can be expanded in large $\delta$, which will turn out to play the role of the spectral parameter.[7]

The local operator $R(\delta)$ is subject to the BRST invariance condition. The BRST variation of the local operator gets contributions from both the line defect and the surface defect. The condition for its BRST-invariance at the quantum level is expected to be given in terms of the coproduct as

$$\Delta_{\mathrm{M2,M5}}(g) R = R \Delta_{\mathrm{M2,M5}}^{\mathrm{op}}(g), \qquad \text{for any } g \in Y(\widehat{\mathfrak{gl}}(1)), \qquad (5.6)$$

and its conjugation by $e^{\delta \rho_{\mathcal{L}}(t_{0,1})}$ for $R(\delta)$ [18]. Here, $\Delta_{\mathrm{M2,M5}} := (\rho_{\mathcal{L}} \otimes \rho_{\mathcal{S}})\Delta : Y(\widehat{\mathfrak{gl}}(1)) \to \mathrm{M2} \,\widehat{\otimes}\, U(\mathrm{M5})$ is the mixed coproduct and $\Delta^{\mathrm{op}}$ is the opposite coproduct of $Y(\widehat{\mathfrak{gl}}(1))$.

Moreover, consider the configuration of multiple line and surface defects. When the same kind of defects approach each other and fuse into a single defect, the intersections should fuse into a single intersection accordingly. At the same time, these defects may be separated from each other by an arbitrarily large distance, in which case they do not interact with each other since the long-distance interaction vanishes (2.11) in the 5d CS theory. The equivalence of the two configurations implies the cluster decomposition of the expectation value of the defect configuration.

The two basic cases of such a fusion are 1) a single line defect $\mathcal{L}$ intersecting with two surface defects $\mathcal{S}$ and $\mathcal{S}'$; 2) two line defects $\mathcal{L}$ and $\mathcal{L}'$ intersecting with a single surface defect $\mathcal{S}$. The cluster decomposition leads to

$$R_{\mathcal{L}, \mathcal{S} \circ \mathcal{S}'} = R_{\mathcal{L}, \mathcal{S}} R_{\mathcal{L}, \mathcal{S}'} \qquad (5.7\text{a})$$

$$R_{\mathcal{L} \circ \mathcal{L}', \mathcal{S}} = R_{\mathcal{L}', \mathcal{S}} R_{\mathcal{L}, \mathcal{S}}, \qquad (5.7\text{b})$$

and their conjugation by $e^{\delta \rho_{\mathcal{L}}(t_{0,1})}$ for $R_{\mathcal{L}, \mathcal{S}}(\delta)$. These fusion rules are compatible with the

---

[6]This is not the usual shift automorphism of $Y(\widehat{\mathfrak{gl}}(1))$. See appendix A.4.

[7]Recall that, in the 4d CS theory, the location of the line defect on the holomorphic plane is identified with the spectral parameter for the representation of quantum algebra assigned to the defect [20, 24]. Similarly, in our 5d CS theory setting, the locations of the line defect and the surface defect on the transverse holomorphic plane play the role of the spectral parameter.

BRST invariance condition (5.6). Indeed, it is straightforward to see that

$$
\begin{aligned}
\Delta(g)R_{\mathcal{L},\mathcal{S}\circ\mathcal{S}'} &= ((\rho_\mathcal{L}\otimes\rho_\mathcal{S}\otimes\rho_{\mathcal{S}'})(\mathrm{id}\otimes\Delta)\Delta(g))\,R_{\mathcal{L},\mathcal{S}}R_{\mathcal{L},\mathcal{S}'} \\
&= (\rho_\mathcal{L}\otimes\rho_\mathcal{S}\otimes\rho_{\mathcal{S}'})(g_{(1)}\otimes g_{(2),(1)}\otimes g_{(2),(2)})R_{\mathcal{L},\mathcal{S}}R_{\mathcal{L},\mathcal{S}'} \\
&= (\rho_\mathcal{L}\otimes\rho_\mathcal{S}\otimes\rho_{\mathcal{S}'})(g_{(1),(1)}\otimes g_{(1),(2)}\otimes g_{(2)})R_{\mathcal{L},\mathcal{S}}R_{\mathcal{L},\mathcal{S}'} \\
&= R_{\mathcal{L},\mathcal{S}}\left((\rho_\mathcal{L}\otimes\rho_\mathcal{S}\otimes\rho_{\mathcal{S}'})(g_{(1),(2)}\otimes g_{(1),(1)}\otimes g_{(2)})\right)R_{\mathcal{L},\mathcal{S}'} \\
&= R_{\mathcal{L},\mathcal{S}}\left((\rho_\mathcal{L}\otimes\rho_\mathcal{S}\otimes\rho_{\mathcal{S}'})(g_{(2),(1)}\otimes g_{(1)}\otimes g_{(2),(2)})\right)R_{\mathcal{L},\mathcal{S}'} \\
&= R_{\mathcal{L},\mathcal{S}}R_{\mathcal{L},\mathcal{S}'}(\rho_\mathcal{L}\otimes\rho_\mathcal{S}\otimes\rho_{\mathcal{S}'})(g_{(2),(2)}\otimes g_{(1)}\otimes g_{(2),(1)}) \\
&= R_{\mathcal{L},\mathcal{S}}R_{\mathcal{L},\mathcal{S}'}(\rho_\mathcal{L}\otimes\rho_\mathcal{S}\otimes\rho_{\mathcal{S}'})(g_{(2)}\otimes g_{(1),(1)}\otimes g_{(1),(2)}) \\
&= R_{\mathcal{L},\mathcal{S}\circ\mathcal{S}'}\Delta^{\mathrm{op}}(g),
\end{aligned}
\tag{5.8}
$$

where we used coassociativity $(\Delta\otimes\mathrm{id})\Delta = (\mathrm{id}\otimes\Delta)\Delta$ for the third, fifth, and seventh equalities; and (5.6) for the fourth and sixth equalities. We used the Sweedler's notation $\Delta(g) = g_{(1)}\otimes g_{(2)}$ for the coproduct, omitting the summation symbol. In a similar way, it is easy to show that the fusion (5.7b) is compatible with the BRST invariance condition (5.6). It is important to note that the compatibility holds even if we use the meromorphic coproduct $\Delta(z)$ (4.2) for the fusion of the line defects, since the meromorphic coproduct $\Delta(z)$ is obtained simply by shifting the formal variables and expanding in this shift for the usual coproduct $\Delta$ (see appendix A).

If there exists a universal R-matrix for the affine Yangian of $\mathfrak{gl}(1)$, the properties (5.6) and (5.7) would directly be compared with the properties of the universal R-matrix mapped through the associated representations (see [39] for instance). The existence of the universal R-matrix is not proven as of yet (see [39] for the case of higher-rank affine Yangian). However, we will soon introduce Miura operators as solutions to these constraints, and invoke the intertwining relations between their products with different orderings. This relation is identified as the Yang-Baxter equation among the representations of $Y(\widehat{\mathfrak{gl}}(1))$ assigned to M2- and M5-branes. Thus, we call $R(\delta)$ an R-matrix in this sense.

In our perturbative analysis of the 5d CS theory, we will confirm that the intersection of line defect and surface defect indeed gives rise to such an R-matrix by computing the expectation value of the intersecting defect configuration, up to the following two restrictions:

- The 5d CS theory is defined on $\mathbb{R}_t\times\mathbb{C}_x\times\mathbb{C}_z$, not on $\mathbb{R}_t\times\mathbb{C}_x\times\mathbb{C}_z^\times$, in our computation. The mode expansion of the ghost truncates as a result, so that the universal associative algebra is the 1-shifted affine Yangian, $Y_1(\widehat{\mathfrak{gl}}(1))$ (which is a subalgebra of $Y(\widehat{\mathfrak{gl}}(1))$). Moreover, the effect of capping the support of the surface defect $\mathbb{C}^\times$ to $\mathbb{C}$ turns out to be providing a vacuum of the M5-brane algebra acted on by the R-matrix. Note that when we restrict to $g\in Y_1(\widehat{\mathfrak{gl}}(1))\subset Y(\widehat{\mathfrak{gl}}(1))$, for $\Delta^{\mathrm{op}}(g) = g_{(2)}\otimes g_{(1)}$, we have $g_{(1)}\in Y_1(\widehat{\mathfrak{gl}}(1))$ which annihilates the vacuum $|\varnothing\rangle$ unless $g_{(1)} = \mathrm{id}$ (see appendix A.2,

in particular equation (A.14)). In this case, $g_{(2)} = g$ so that the BRST invariance condition for the R-matrix acting on the vacuum reads[8]

$$\Delta_{\text{M2,M5}}(g)R|\varnothing\rangle = R|\varnothing\rangle\rho_{\text{M2}}(g), \qquad \text{for any } g \in Y_1(\widehat{\mathfrak{gl}}(1)), \tag{5.9}$$

and its conjugation by $e^{\delta(t_{0,1}\otimes\text{id})}$ for $R(\delta)$. This will be the object that we actually compute as the expectation value of the defect configuration in the 5d CS theory.

- As we explained in section 2.3, we will only consider the surface defects which originate from parallel M5-branes supported on $\mathbb{R}^2_{\varepsilon_2} \times \mathbb{R}^2_{\varepsilon_3} \times \mathbb{C}_z$; namely, the one of type $\mathcal{S}_{N,0,0}$. Upon the reduction to the IIA theory, these M5-branes descend to the D4-branes which intersect the emergent D6-brane transversally in the topological part of the worldvolume. The back-reaction of these D4-branes modifies the perturbative analysis of the 5d CS theory by sourcing a singularity of the fields at the locus of the surface defect [14].

With these two restrictions understood, we will first introduce Miura operators as solutions to the constraint (5.6) and (5.7). Then, we will compute the expectation value of the intersection of line defect and surface defect, and show they reproduce the Miura operators and their products mapped through the relevant representations.

## 5.1 Miura operators as R-matrices

The solutions to the constraint (5.6) are given by the Miura operators and their products [36, Theorem 16], which are (pseudo-)differential operators providing free field realization of the $Y$-algebra [15, 17]. Let us first consider the Miura operators which are solutions in the cases of a single M2-brane and a single M5-brane. According to the relative orientation of the M2-brane and the M5-brane, there are two possibilities.

First, the M2-brane and the M5-brane can be totally transverse to each other. Due to the triality, we may restrict to the case with the M2-brane supported on $\mathbb{R}^2_{\varepsilon_1} \times \mathbb{R}_t$ and the M5-brane is supported on $\mathbb{R}^2_{\varepsilon_2} \times \mathbb{R}^2_{\varepsilon_3} \times \mathbb{C}_z^\times$. The solution to the constraint (5.6) is given by a differential operator,

$$R_{\text{M2}_{1,0,0},\text{M5}_{1,0,0}} = \varepsilon_1\partial_z - \varepsilon_2\varepsilon_3 J(z) = \varepsilon_1(\partial_z \otimes 1) - \varepsilon_2\varepsilon_3\sum_{n\in\mathbb{Z}} z^n \otimes J_{-n-1}, \tag{5.10}$$

called the Miura operator. Here, the differential operator $\partial_z$ and the Laurent polynomial $z^n$ are viewed as generators of $\text{M2}_{1,0,0}$, so that $R_{\text{M2}_{1,0,0},\text{M5}_{1,0,0}}$ is an element in $\text{M2}_{1,0,0} \widehat{\otimes} U(\text{M5}_{1,0,0})$.

As discussed earlier, the line defect may be translated to $\delta \in \mathbb{C}_x$ along the holomorphic plane transverse to the surface defect. As a result, the Miura operator is conjugated by $e^{\delta \rho_{\text{M2}_{1,0,0}}(t_{0,1})} = e^{\frac{\delta z}{\varepsilon_1}}$. Expanding the result in large $\delta$, we obtain

$$R_{\text{M2}_{1,0,0},\text{M5}_{1,0,0}}(\delta) = 1 + \frac{1}{\delta}\left(\varepsilon_1(\partial_z \otimes 1) - \varepsilon_2\varepsilon_3\sum_{n\in\mathbb{Z}} z^n \otimes J_{-n-1}\right). \tag{5.11}$$

---

[8]We use the abbreviation $R|\varnothing\rangle \equiv R(1 \otimes |\varnothing\rangle)$.

This is the expression that will be compared with the expectation value of the intersecting defects in the 5d CS theory.

Second, the M2-brane and the M5-brane may share a topological plane in their support. Due to the triality, it is enough to consider the M2-brane supported on $\mathbb{R}^2_{\varepsilon_3} \times \mathbb{R}_t$ and the M5-brane supported on $\mathbb{R}^2_{\varepsilon_2} \times \mathbb{R}^2_{\varepsilon_3} \times \mathbb{C}^\times_z$. The solution to the constraint (5.6) in this case is given by a pseudo-differential operator,

$$
\begin{aligned}
R_{\mathrm{M2}_{0,0,1},\mathrm{M5}_{1,0,0}} &= (\varepsilon_3 \partial_z)^{\frac{\varepsilon_1}{\varepsilon_3}} + \sum_{j=1}^\infty U_j(z)(\varepsilon_3 \partial_z)^{\frac{\varepsilon_1}{\varepsilon_3}-j} \\
&= \left( 1 - \varepsilon_1 \varepsilon_2 J(z)(\varepsilon_3 \partial_z)^{-1} + \frac{\varepsilon_1 \varepsilon_2 (\varepsilon_1 - \varepsilon_3)}{2}(\varepsilon_2 J(z)^2 - \partial J(z))(\varepsilon_3 \partial_z)^{-2} + \cdots \right)(\varepsilon_3 \partial_z)^{\frac{\varepsilon_1}{\varepsilon_3}},
\end{aligned}
\tag{5.12}
$$

which is also called the Miura operator. Here, $U_j(z)$ is a spin-$j$ current given in terms of the spin-1 current $J(z)$ and its derivatives, whose generic expression is given by [17]

$$
U_j(z) = \varepsilon_1^j \prod_{k=1}^{j-1}\left(1 - \frac{k\varepsilon_3}{\varepsilon_1}\right) \sum_{m_1+2m_2+\cdots+jm_j=j} \prod_{k=1}^j \frac{1}{m_k! k^{m_k}}\left(\frac{-\varepsilon_2}{(k-1)!}\partial^{k-1}J\right)^{m_k}. \tag{5.13}
$$

This pseudo-differential operator is not explicitly an element in $\mathrm{M2}_{0,0,1} \widehat{\otimes} U(\mathrm{M5}_{1,0,0})$. However, we can translate the line defect to $\delta \in \mathbb{C}_x$, resulting in the conjugation of the Miura operator by $e^{\delta \rho_{\mathrm{M2}_{0,0,1}}(t_{0,1})} = e^{\frac{\delta z}{\varepsilon_3}}$. The first few terms of the large $\delta$ expansion are given by

$$
\begin{aligned}
R_{\mathrm{M2}_{0,0,1},\mathrm{M5}_{1,0,0}}(\delta) = 1 &+ \frac{1}{\delta}\left( \varepsilon_1(\partial_z \otimes 1) - \varepsilon_1 \varepsilon_2 \sum_{n\in\mathbb{Z}} z^n \otimes J_{-n-1} \right) \\
&+ \frac{1}{\delta^2}\left( \frac{\varepsilon_1(\varepsilon_1 - \varepsilon_3)}{2}(\partial_z^2 \otimes 1) - \varepsilon_1 \varepsilon_2(\varepsilon_1 - \varepsilon_3)\sum_{n\in\mathbb{Z}} z^n \partial_z \otimes J_{-n-1} \right. \\
&\left. \quad + \frac{\varepsilon_1 \varepsilon_2(\varepsilon_1 - \varepsilon_3)}{2}\left( \varepsilon_2 \sum_{m,n\in\mathbb{Z}} z^{m+n} \otimes J_{-m-1}J_{-n-1} - \sum_{n\in\mathbb{Z}}(n+1)z^n \otimes J_{-n-2} \right) \right) \\
&+ \mathcal{O}(\delta^{-3}),
\end{aligned}
\tag{5.14}
$$

where we discarded the unimportant normalization factor $\delta^{\frac{\varepsilon_1}{\varepsilon_3}}$. Note that each coefficient in this $\delta^{-1}$-expansion is indeed valued in $\mathrm{M2}_{0,0,1} \widehat{\otimes} U(\mathrm{M5}_{1,0,0})$. This is the object that we will reproduce as the expectation value of the intersecting defect configuration in the 5d CS theory.

We proceed to intersections of more complicated line defect and surface defect, constructed by multiple M2-branes and M5-branes. Such intersections can always be reconstructed by fusing the basic intersections of a single M2-brane and a single M5-brane that we

just studied, through the fusion rules (5.7). Therefore, the local operator $R$ at the intersection is always given as a product of the Miura operators (5.10) and (5.12). When the line defect is translated to $\delta \in \mathbb{C}_x$, we only need to conjugate this product with $e^{\delta \rho_{\mathcal{L}}(t_{0,1})}$ to obtain $R(\delta)$.

When the defect configuration is composed of a single M2-brane and multiple M5-branes, their intersection establishes the Miura transformation for the $Y$-algebra [15],

$$R_{\text{M2}_{1,0,0},\text{M5}_{L,M,N}} = R_{\text{M2}_{1,0,0},\text{M5}_{c_1}} R_{\text{M2}_{1,0,0},\text{M5}_{c_2}} \cdots R_{\text{M2}_{1,0,0},\text{M5}_{c_{L+M+N}}}, \tag{5.15}$$

where we used $c : \{1, 2, \cdots, L + M + N\} \to \{1, 2, 3\}$ to denote the orientations of the M5-branes and abbreviated $c_I \equiv (\delta_{a,c_I})_{a=1,2,3}$. When the ordering of any two Miura operators is exchanged, the Miura transformation should still generates an isomorphic $Y$-algebra. This isomorphism is established by the Maulik-Okounkov R-matrix (and its generalizations incorporating non-parallel M5-branes [40]), $R_{\text{M5}_{c_I},\text{M5}_{c_{I+1}}} \in U(\text{M5}_{c_I}) \widehat{\otimes} U(\text{M5}_{c_{I+1}})$, which relates the products of Miura operators with different orderings by [9]

$$R_{\text{M2}_{1,0,0},\text{M5}_{c_I}} R_{\text{M2}_{1,0,0},\text{M5}_{c_{I+1}}} R_{\text{M5}_{c_I},\text{M5}_{c_{I+1}}} = R_{\text{M5}_{c_I},\text{M5}_{c_{I+1}}} R_{\text{M2}_{1,0,0},\text{M5}_{c_{I+1}}} R_{\text{M2}_{1,0,0},\text{M5}_{c_I}}. \tag{5.16}$$

As we reconstructed the Miura operators in terms of the representations of $Y(\widehat{\mathfrak{gl}}(1))$, the above relation is identified as the Yang-Baxter equation for the relevant representations of $Y(\widehat{\mathfrak{gl}}(1))$. In this sense, we call the Miura operators and their products R-matrices of the affine Yangian of $\mathfrak{gl}(1)$.[9]

## 5.2 Expectation value of intersection of line defect and surface defect

Let us compute the expectation value of the intersection of a line defect and a surface defect. We take a generic line defect $\mathcal{L}$ placed at $(\delta, 0) \in \mathbb{C}_x \times \mathbb{C}_z$ and the surface defect $\mathcal{S}_{N,0,0}$ descending from $N$ M5-branes wrapping $\mathbb{R}^2_{\varepsilon_2} \times \mathbb{R}^2_{\varepsilon_3} \times \mathbb{C}_z$, located at the origin of the transverse holomorphic plane $0 \in \mathbb{C}_x$. Later we shall compare this expectation value with the output of the R-matrix acting on a certain vacua. Note that the two defects are separated by a distance $\delta$, which will play the role of the *spectral parameter* for the R-matrix. We compute the vacuum expectation value of the intersecting defect configuration as a series in $\delta^{-1}$.

Two basic diagrams to compute are:

$$G_{t \otimes W} := \tag{5.17}$$

[9]In the multiplicative uplift realized by the twisted M-theory on $\mathbb{R}^2_{\varepsilon_1} \times \mathbb{R}^2_{\varepsilon_2} \times \mathbb{R}^2_{\varepsilon_3} \times \mathbb{R}_t \times \mathbb{C}^\times_X \times \mathbb{C}^\times_Z$, it was suggested that the universal associative algebra is the quantum toroidal algebra of $\mathfrak{gl}(1)$ [18], which does possess a universal R-matrix. There, the Miura transformation of the $q$-deformed $W$- and $Y$-algebra was established precisely by the R-matrices for the representations assigned to the M2-branes and the M5-branes.

$$= \alpha_s \alpha_l \sum_{m,n,p,q=0}^{\infty} \frac{t_{m,n}}{m!n!p!q!} \int_{\mathbb{R}} \mathrm{d}t \int_{\mathbb{C}} \mathrm{d}^2 z_2 \partial_{x_1}^m \partial_{z_1}^n \overbracket{A_t(v_1) \partial_{x_2}^p A_{\bar{z}}(v_2)} z_2^q \partial_{z_2}^q W^{(p+1)}(z_2) \Big|_{\substack{x_1=\delta \\ z_1,x_2=0}}$$

where $v_1 = (t, x_1, \bar{x}_1, z_1, \bar{z}_1)$, $v_2 = (0, x_2, \bar{x}_2, z_2, \bar{z}_2)$, and

$$G_{t \otimes 1} :=$$

$$= \alpha_l \sum_{m,n=0}^{\infty} \frac{t_{m,n}}{m!n!} \int_{\mathbb{R}} \mathrm{d}t \partial_x^m \partial_z^n A_t^{(0)}(v) \Big|_{\substack{x=\delta \\ z=0}} \qquad (5.18)$$

where $v = (t, x, \bar{x}, z, \bar{z})$. One can check that up to two loop order (quadratic in $\varepsilon_1$) there are no more connected diagrams that contribute non-trivially.[10] There are some disconnected diagrams that we shall get to momentarily.

We note that, in (5.17), we can replace the derivative of the current with one of its modes, namely $\frac{1}{q!} \partial_z^q W^{(p+1)}(z) \big|_{z=0} = W_{-p-q-1}^{(p+1)}$, and we substitute the expression for the correlation function (2.15) – to find:

$$G_{t \otimes W} = \frac{3\varepsilon_1 \alpha_s \alpha_l}{32\pi^2} \sum_{m,n,p,q=0}^{\infty} \frac{t_{m,n}}{m!n!p!} \int_{\mathbb{R}} \mathrm{d}t \int_{\mathbb{C}} \mathrm{d}^2 z_2 \partial_{x_1}^m \partial_{z_1}^n \partial_{x_2}^p \frac{-2\bar{x}_{12} z_2^q W_{-p-q-1}^{(p+1)}}{(t^2 + |x_{12}|^2 + |z_{12}|^2)^{5/2}} \Big|_{\substack{x_1=\delta \\ x_2,z_1=0}}$$

$$= \frac{3\varepsilon_1 \alpha_s \alpha_l}{16\pi^2} \sum_{m,n,p,q=0}^{\infty} \frac{t_{m,n} \otimes W_{-p-q-1}^{(p+1)}}{m!n!p!} (-1)^{n+p+1} \frac{\Gamma\left(-\frac{3}{2}\right)}{\Gamma\left(-\frac{3}{2} - m - n - p\right)}$$

$$\times \int_{\mathbb{R}} \mathrm{d}t \int_{\mathbb{C}} \mathrm{d}^2 z \frac{\bar{\delta}^{m+p+1} \bar{z}^n z^q}{(t^2 + |\delta|^2 + |z|^2)^{5/2 + m + n + p}}$$

$$= \frac{\mathrm{i}}{2\pi} \varepsilon_1 \alpha_s \alpha_l \sum_{m,n,p=0}^{\infty} \frac{(-1)^m}{\delta^{m+p+1}} \binom{m+p}{m} t_{m,n} \otimes W_{-p-n-1}^{(p+1)} \qquad (5.19)$$

In (5.18), $A^{(0)}$ is $z$-independent and so only the $n = 0$ term contributes. Furthermore, the $m = 0$ term contributes a logarithmically (in $\delta$) divergent term which we can cancel by a counterterm. Let us denote this renormalized diagram by $G_{t \otimes 1}^{\mathrm{ren}}$. Replacing $\partial_x A_t$ with $F_{xt}$ from (2.26) we get:

$$G_{t \otimes 1}^{\mathrm{ren}} = -\frac{N}{4} \frac{\varepsilon_1}{\varepsilon} \alpha_l \sum_{m=1}^{\infty} \frac{t_{m,0}}{m!} \int_{\mathbb{R}} \mathrm{d}t \partial_x^{m-1} \frac{2\bar{x}}{(t^2 + |x|^2)^{3/2}} \Big|_{x=\delta}$$

$$= -\frac{N}{2} \frac{\varepsilon_1}{\varepsilon} \alpha_l \sum_{m=1}^{\infty} \frac{t_{m,0}}{m!} \frac{\Gamma\left(-\frac{1}{2}\right)}{\Gamma\left(\frac{1}{2} - m\right)} \int_{\mathbb{R}} \mathrm{d}t \frac{\bar{\delta}^m}{(t^2 + |\delta|^2)^{1/2 + m}}$$

---

[10] Any connected 2-loop candidate diagram would necessarily contain on of the vanishing subdiagrams of Appendix C and therefore would vanish itself.

$$= N\frac{\varepsilon_1}{\varepsilon}\alpha_l \sum_{m=1}^{\infty} \frac{(-1)^m}{m}\frac{t_{m,0}}{\delta^m} \tag{5.20}$$

In addition to these two connected diagrams, there are three disconnected diagrams we need to compute:

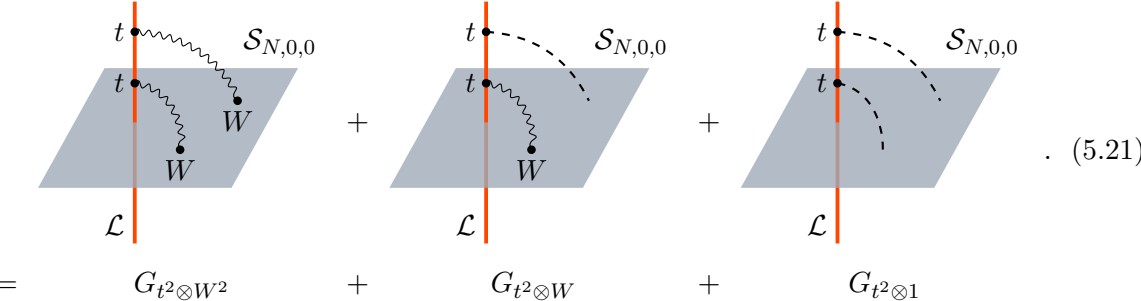

$$= \qquad G_{t^2\otimes W^2} \qquad + \qquad G_{t^2\otimes W} \qquad + \qquad G_{t^2\otimes 1} \qquad . \tag{5.21}$$

These diagrams are roughly given by the products of their connected components[11] and given that we have already computed the connected pieces (5.19) and (5.20), this may seem straightforward. The subtlety is that we have to use path ordering for the $t_{m,n}$s inserted along the line defect and radial ordering for the mode operators along the surface defect. At a generic order in $\delta^{-1}$ this can get quite complicated, however, here we are only going to be concerned with the R-matrix up to order $\delta^{-2}$.

Both of the connected diagrams, (5.19) and (5.20), start from $\delta^{-1}$ and therefore to evaluate the disconnected diagram up to order $\delta^{-1}$ we only need to consider the products of the leading terms from the connected diagrams. The leading term of (5.20) has the operator $t_{1,0}$ along the line which commutes with itself, so there is no problem with simply squaring the leading term without worrying about path ordering along the line. On the defect we only have the identity operator for this diagram, so we don't need to worry about radial ordering either. Similarly, the leading term of (5.19) contains the operators $t_{0,n}$ which also commute among themselves (B.1 and B.2). Here we also have two modes $W^{(1)}_{-n-1} = -J_{-n-1}$ of the spin-1 current on the surface defect but they are always negative modes and as such commute among themselves. As a result:

$$\lim_{\delta\to\infty} \delta^2 G_{t^2\otimes W^2} = \frac{1}{2}\left(\lim_{\delta\to\infty}\delta G_{t\otimes W}\right)^2 \quad \text{and} \quad \lim_{\delta\to\infty}\delta^2 G_{t^2\otimes 1} = \frac{1}{2}\left(\lim_{\delta\to\infty}\delta G_{t\otimes 1}^{\text{ren}}\right)^2. \tag{5.22}$$

For the remaining diagram we need to consider path ordering since $t_{1,0}$ does not commute with $t_{0,n}$ for nonzero $n$. We don't need to consider radial ordering on the surface defect because there is one mode operator coming from $G_{t\otimes W}$ and only the identity operator from $G_{t\otimes 1}$. The consequence of path ordering is that instead of taking the simple product of the two diagrams we end up taking the *symmetric product*:

$$\lim_{\delta\to\infty}\delta^2 G_{t^2\otimes W} = \text{Sym}\left[\left(\lim_{\delta\to\infty}\delta G_{t\otimes W}\right)\left(\lim_{\delta\to\infty}\delta G_{t\otimes 1}^{\text{ren}}\right)\right] \tag{5.23}$$

---

[11]And symmetry factors. The symmetry factor is $\frac{1}{2}$ for $G_{t^2\otimes W^2}$ and $G_{t^2\otimes 1}$ since we can permute the two propagators or the two background fields. $G_{t^2\otimes W}$ has no such symmetry and so its symmetry factor is 1.

where $\text{Sym}[O_1 O_2] = \frac{1}{2}(O_1 O_2 + O_2 O_1)$ for any two operators $O_1$ and $O_2$.

Summing up all the diagrams, including the trivial diagram without any propagator or background field that gives 1, and substituting the values of the parameters $\alpha_s = \sqrt{\varepsilon_2 \varepsilon_3}$ (3.18, 3.19), $\varepsilon = 2\pi\sqrt{-\varepsilon_2 \varepsilon_3}$ (3.26), $\alpha_l = -2\pi\sqrt{-\varepsilon_2 \varepsilon_3}$ (3.8) that we have fixed by canceling anomalies, we find the expectation value of the intersecting defects:

$$
\begin{aligned}
\langle \mathcal{L} \otimes \mathcal{S}_{N,0,0} \rangle(\delta) &= 1 + G_{t \otimes 1}^{\text{ren}} + G_{t \otimes W} + G_{t^2 \otimes 1} + G_{t^2 \otimes W} + G_{t^2 \otimes W^2} + \mathcal{O}(\varepsilon_1^3) \\
&= 1 + \frac{1}{\delta}\left( N\varepsilon_1 t_{1,0} - \sigma_3 \sum_{n=0}^{\infty} t_{0,n} \otimes J_{-n-1}\right) \\
&\quad + \frac{1}{\delta^2}\left( -\frac{N}{2}\varepsilon_1 t_{2,0} + \sigma_3 \sum_{n=0}^{\infty}(t_{0,n} \otimes L_{-n-2} + t_{1,n} \otimes J_{-n-1}) \right. \\
&\quad + \frac{N^2}{2}\varepsilon_1^2 t_{1,0} t_{1,0} - N\varepsilon_1 \sigma_3 \sum_{n=0}^{\infty} \frac{1}{2}(t_{1,0} t_{0,n} + t_{0,n} t_{1,0}) \otimes J_{-n-1} \\
&\quad \left. + \frac{1}{2}\sigma_3^2 \sum_{m,n=0}^{\infty} t_{0,n} t_{0,m} \otimes J_{-n-1} J_{-m-1}\right) \\
&\quad + \cdots
\end{aligned}
\tag{5.24}
$$

where $\sigma_3 = \varepsilon_1 \varepsilon_2 \varepsilon_3$ and $\cdots$ refers to terms of higher order in $\varepsilon_1$ and/or in $\delta^{-1}$. As usual, we are writing the modes of the spin-1 and spin-2 currents as $J$ and $T$, i.e., $J_n = -W_n^{(1)}$ and $L_n = W_n^{(2)}$. The order-$\delta^{-1}$ terms and the first line of the order-$\delta^{-2}$ terms (inside parentheses) are the leading and the next-to-leading terms in the connected diagrams (5.19) and (5.20). The terms without any $J$-mode and those quadratic in the $J$-modes are from the diagrams $G_{t^2 \otimes 1}$ and $G_{t^2 \otimes W^2}$ respectively and are accompanied by the symmetry factor $\frac{1}{2}$. The remaining term is from the remaining diagram $G_{t^2 \otimes W}$.

Next we shall compare the expectation value (5.24) of intersecting defects with the R-matrix acting on a vacuum.

## 5.3 Miura operators from defect intersections

The R-matrix (acting on the vacuum) (5.24) that we computed in the 5d CS theory is universal in the sense that the line defect and the surface defect were not specified, except the orientation of the M5-branes constructing the surface defect. We now demonstrate that, when the line defect and the surface defect are specified, this R-matrix mapped through the associated representations exactly matches with the Miura operators and their products.

### 5.3.1 Single M2-M5 intersection

Let us consider the simplest defect intersection, where the line defect descends from a single M2-brane and the surface defect descends from a single M5-brane. We distinguish the two cases: 1) the transverse intersection between $\mathcal{L}_{1,0,0}$ and $\mathcal{S}_{1,0,0}$; 2) the non-transverse intersection between $\mathcal{L}_{0,0,1}$ and $\mathcal{S}_{1,0,0}$. The non-transverse intersection between $\mathcal{L}_{0,1,0}$ and $\mathcal{S}_{1,0,0}$ is

obtained from the latter case simply by the symmetry of exchanging $\mathbb{R}^2_{\varepsilon_2}$ and $\mathbb{R}^2_{\varepsilon_3}$.

We begin with the transverse intersection of $\mathcal{L}_{1,0,0}$ and $\mathcal{S}_{1,0,0}$, for which the expectation value (5.24) is represented on $\mathrm{M2}_{1,0,0} \widehat{\otimes} U(\mathrm{M5}_{1,0,0})$. After substituting the generators mapped through the associated representations, we get

$$\langle \mathcal{L}_{1,0,0} \otimes \mathcal{S}_{1,0,0} \rangle(\delta) = 1 + \frac{1}{\delta} \left( \varepsilon_1 \partial_z \otimes 1 - \varepsilon_2 \varepsilon_3 \sum_{n=0}^{\infty} z^n \otimes J_{-n-1} \right), \tag{5.25}$$

where we used the generators (2.21) for the single M2-brane and the Sugawara tensor

$$T(z) = -\frac{\varepsilon_2 \varepsilon_3}{2}(JJ)(z), \tag{5.26}$$

for the single M5-brane. It is important to note the $\delta^{-2}$-order term vanishes. We expect that the terms higher-order in $\delta^{-1}$ also vanish when represented on $\mathrm{M2}_{1,0,0} \widehat{\otimes} U(\mathrm{M5}_{1,0,0})$, so that the above expression is exact. This precisely recovers the Miura operator (5.11) acting on the vacuum $|\varnothing\rangle$.

Next, we consider the non-transverse intersection of $\mathcal{L}_{0,0,1}$ and $\mathcal{S}_{1,0,0}$. As in the previous case, we can represent the expectation value (5.24) on $\mathrm{M2}_{0,0,1} \widehat{\otimes} U(\mathrm{M5}_{1,0,0})$. Note that the M2-brane with a different orientation results in switching the $\Omega$-background parameter appearing in the differential operators. The outcome is

$$
\begin{aligned}
\langle \mathcal{L}_{0,0,1} &\otimes \mathcal{S}_{1,0,0} \rangle(\delta) \\
&= 1 + \frac{1}{\delta} \left( \varepsilon_1 (\partial_z \otimes 1) - \varepsilon_1 \varepsilon_2 \sum_{n=0}^{\infty} z^n \otimes J_{-n-1} \right) \\
&\quad + \frac{1}{\delta^2} \left( \frac{\varepsilon_1(\varepsilon_1 - \varepsilon_3)}{2} (\partial_z^2 \otimes 1) - \varepsilon_1 \varepsilon_2(\varepsilon_1 - \varepsilon_3) \sum_{n=0}^{\infty} z^n \partial_z \otimes J_{-n-1} \right. \\
&\quad \left. + \frac{\varepsilon_1 \varepsilon_2(\varepsilon_1 - \varepsilon_3)}{2} \left( \varepsilon_2 \sum_{m,n=0}^{\infty} z^{m+n} \otimes J_{-m-1}J_{-n-1} - \sum_{n=0}^{\infty}(n+1)z^n \otimes J_{-n-2} \right) \right) \\
&\quad + \mathcal{O}(\delta^{-3}),
\end{aligned} \tag{5.27}
$$

which exactly matches with the Miura operator (5.14) for the non-transverse intersection acting on the vacuum, up to the $\delta^{-2}$-order.

### 5.3.2 One M2-brane and two M5-branes

Next, we take the next-to-simplest transverse intersection where the line defect $\mathcal{L}_{1,0,0}$ is still given by a single M2-brane but the surface defect $\mathcal{S}_{2,0,0}$ is engineered from two parallel M5-

branes. When represented on $\mathrm{M2}_{1,0,0} \widehat{\otimes} U(\mathrm{M5}_{2,0,0})$, the expectation value (5.24) becomes

$$\langle \mathcal{L}_{1,0,0} \otimes \mathcal{S}_{2,0,0} \rangle(\delta) = 1 + \frac{1}{\delta}\left(2\varepsilon_1\partial_z - \varepsilon_2\varepsilon_3 \sum_{n=0}^{\infty} z^n \otimes J_{-n-1}\right)$$

$$+ \frac{1}{\delta^2}\left[\varepsilon_1^2\partial_z^2 - \sigma_3 \sum_{n=0}^{\infty} z^n \partial_z \otimes J_{-n-1} + \frac{\varepsilon_2^2\varepsilon_3^2}{2} \sum_{m,n=0}^{\infty} z^{m+n} \otimes J_{-m-1}J_{-n-1}\right.$$

$$\left. - \frac{\sigma_3}{2} \sum_{n=0}^{\infty} (n+1)z^n \otimes J_{-n-2} + \varepsilon_2\varepsilon_3 \sum_{n=0}^{\infty} z^n \otimes L_{-n-2}\right], \tag{5.28}$$

by using (2.21). We expect that the terms higher-order in $\delta^{-1}$ vanish, so that what is obtained here is the exact expression.

Meanwhile, by (5.7a), the Miura transformation for the vertex algebra on the surface defect $\mathcal{S}_{2,0,0}$ (i.e., the $\mathcal{W}_2$-algebra) is realized by the product of two Miura operators,

$$R_{\mathrm{M2}_{1,0,0},\mathrm{M5}_{2,0,0}} = (\varepsilon_1\partial_z - \varepsilon_2\varepsilon_3 J_1(z))\,(\varepsilon_1\partial_z - \varepsilon_2\varepsilon_3 J_2(z))$$

$$= \varepsilon_1^2\partial_z^2 - \sigma_3\,(J_1(z) + J_2(z))\,\partial_z + \varepsilon_2^2\varepsilon_3^2 J_1(z)J_2(z) - \sigma_3\partial J_2(z), \tag{5.29}$$

where we remind that $J_1(z) = \sum_{n\in\mathbb{Z}}(J_1)_{-n-1}z^n = \varepsilon_1 \sum_{n\in\mathbb{Z}} t_{0,n} \otimes J_{-n-1} \otimes 1$ which is regarded as an element in $\mathrm{M2}_{1,0,0} \widehat{\otimes} U(\mathrm{M5}_{1,0,0}) \widehat{\otimes} U(\mathrm{M5}_{1,0,0})$, and similarly for $J_2(z)$. We recall that the generating currents of the $\mathcal{W}_2$-algebra $\mathrm{M5}_{2,0,0}$ in the primary basis are expressed as

$$J(z) = J_1(z) + J_2(z)$$

$$T(z) = -\frac{\varepsilon_2\varepsilon_3}{2}\left(J_1(z)^2 + J_2(z)^2\right) + \frac{\varepsilon_1}{2}\left(\partial J_1(z) - \partial J_2(z)\right), \tag{5.30}$$

under this Miura transformation realizing an embedding $\mathrm{M5}_{2,0,0} \hookrightarrow \mathrm{M5}_{1,0,0} \otimes \mathrm{M5}_{1,0,0}$.

In terms of these generating currents, the Miura transformation (5.29) is organized into

$$R_{\mathrm{M2}_{1,0,0},\mathrm{M5}_{2,0,0}} = \varepsilon_1^2\partial_z^2 - \sigma_3 J(z)\partial_z + \frac{\varepsilon_2^2\varepsilon_3^2}{2}J(z)^2 - \frac{\sigma_3}{2}\partial J(z) + \varepsilon_2\varepsilon_3 T(z). \tag{5.31}$$

Acting the vacuum $|\varnothing\rangle$ of the $\mathcal{W}_2$-algebra from the right, we get

$$R_{\mathrm{M2}_{1,0,0},\mathrm{M5}_{2,0,0}}|\varnothing\rangle = \left[\varepsilon_1^2\partial_z^2 - \sigma_3 \sum_{n=0}^{\infty} z^n \partial_z \otimes J_{-n-1} + \frac{\varepsilon_2^2\varepsilon_3^2}{2} \sum_{m,n=0}^{\infty} z^{m+n} \otimes J_{-m-1}J_{-n-1}\right.$$

$$\left. - \frac{\sigma_3}{2} \sum_{n=0}^{\infty} (n+1)z^n \otimes J_{-n-2} + \varepsilon_2\varepsilon_3 \sum_{n=0}^{\infty} z^n \otimes L_{-n-2}\right]|\varnothing\rangle. \tag{5.32}$$

We confirm that the expectation value (5.28) of the defect intersection in the 5d CS theory exactly matches with this Miura transformation acting on the vacuum, after conjugating the latter with $e^{\delta\,\rho_{\mathrm{M2}_{1,0,0}}(t_{0,1})} = e^{\frac{\delta z}{\varepsilon_1}}$ and expanding it in $\delta^{-1}$.

### 5.3.3   Two M2-branes and one M5-brane

Finally, let us consider the case where the line defect $\mathcal{L}_{2,0,0}$ constructed by two parallel M2-branes and the surface defect $\mathcal{S}_{1,0,0}$ constructed by a single M5-brane form a transverse intersection. We represent the R-matrix (5.24) on $\mathrm{M2}_{2,0,0} \widehat{\otimes} U(\mathrm{M5}_{1,0,0})$ to get

$$
\begin{aligned}
\langle \mathcal{L}_{2,0,0} \otimes \mathcal{S}_{1,0,0} \rangle(\delta) = 1 &+ \frac{1}{\delta}\left( \varepsilon_1(\partial_{z_1} + \partial_{z_2}) - \varepsilon_2\varepsilon_3 \sum_{n=0}^{\infty}(z_1^n + z_2^n) \otimes J_{-n-1} \right) \\
&+ \frac{1}{\delta^2}\left( \left( \varepsilon_1^2 \partial_{z_1}\partial_{z_2} - \frac{\varepsilon_2\varepsilon_3}{(z_1 - z_2)^2} \right) \otimes 1 - \sigma_3 \sum_{n=0}^{\infty}(z_1^n \partial_{z_2} + z_2^n \partial_{z_1}) \otimes J_{-n-1} \right. \\
&\left. \quad\quad + \frac{\varepsilon_2^2\varepsilon_3^2}{2} \sum_{m,n=0}^{\infty}(z_1^m z_2^n + z_1^n z_2^m) \otimes J_{-m-1}J_{-n-1} \right),
\end{aligned}
\tag{5.33}
$$

by using (2.22) and (5.26). We expect that the terms higher-order in $\delta^{-1}$ vanish, so that the obtained expression is in fact exact.

Meanwhile, for the transverse intersections of two M2-branes and a single M5-brane, the fusion rule (5.7b) indicates that the R-matrix is given by the product of two Miura operators,

$$
\begin{aligned}
R_{\mathrm{M2}_{2,0,0},\mathrm{M5}_{1,0,0}} &= (\varepsilon_1\partial_{z_2} - \varepsilon_2\varepsilon_3 J(z_2))\,(\varepsilon_1\partial_{z_1} - \varepsilon_2\varepsilon_3 J(z_1)) \\
&= \left( \varepsilon_1^2 \partial_{z_1}\partial_{z_2} - \frac{\varepsilon_2\varepsilon_3}{(z_1-z_2)^2} \right) - \sigma_3\left( J(z_1)\partial_{z_2} + J(z_2)\partial_{z_1} \right) + \varepsilon_2^2\varepsilon_3^2 : J(z_1)J(z_2) :,
\end{aligned}
\tag{5.34}
$$

where the $\widehat{\mathfrak{gl}}(1)$ currents are normal-ordered to produce a meromorphic function. When acted on the vacuum, it yields

$$
\begin{aligned}
R_{\mathrm{M2}_{2,0,0},\mathrm{M5}_{1,0,0}}|\varnothing\rangle = &\left[ \left( \varepsilon_1^2 \partial_{z_1}\partial_{z_2} - \frac{\varepsilon_2\varepsilon_3}{(z_1-z_2)^2} \right) \otimes 1 - \sigma_3 \sum_{n=0}^{\infty}(z_1^n \partial_{z_2} + z_2^n \partial_{z_1}) \otimes J_{-n-1} \right. \\
&\left. + \frac{\varepsilon_2^2\varepsilon_3^2}{2} \sum_{m,n=0}^{\infty}(z_1^m z_2^n + z_1^n z_2^m) \otimes J_{-m-1}J_{-n-1} \right]|\varnothing\rangle,
\end{aligned}
\tag{5.35}
$$

where we used a symmetric polynomial in $z_1$ and $z_2$ in the last term to put it into an element in $\mathrm{M2}_{2,0,0}$. It is straightforward to see that the expectation value (5.33) of the defect intersection precisely matches with this R-matrix acting on the vacuum, conjugated with $e^{\delta \rho_{\mathrm{M2}_{2,0,0}}(t_{0,1})} = e^{\frac{\delta(z_1+z_2)}{\varepsilon_1}}$ and expanded in $\delta^{-1}$.

## 6   Discussion

There are many interesting directions that deserve further studies.

**Surface defects with different orientations.** We have only discussed surface defects wrapping either $\mathbb{C}_x$ or $\mathbb{C}_z$ of the 5d space-time $\mathbb{R} \times \mathbb{C}_x \times \mathbb{C}_z$. This way, if we have two parallel surface defects and a topological line defect intersecting both, we naturally get a product of *two* R-matrices. The surface defects themselves do not intersect. In an analogous configuration of three line operators in 4d CS theory on $\mathbb{R}^2 \times \mathbb{C}$, the operators can be arranged arbitrarily in the topological plane so that they pairwise intersect and we get the product of *three* R-matrices. Moving the line operators then result in permuting the R-matrices while keeping the overall expectation value invariant – leading to the the Yang-Baxter equation for Yangians [24]. To get the Yang-Baxter equation for the affine Yangians studied in this paper we need surface defects wrapping arbitrary holomorphic curves inside $\mathbb{C}_x \times \mathbb{C}_z$ so that they can intersect. Defining surface defects with these more general supports, computing the expectation values of their intersections, and figuring out whether we can derive the relevant Yang-Baxter equations (1.2) involving the Maulik-Okounkov R-matrix from perturbation theory remains an interesting open direction.

**Higher-rank generalization.** It would be interesting to extend our direct perturbative study to the 5d non-commutative $\mathfrak{gl}(K)$ Chern-Simons theory. In the context of twisted M-theory, this corresponds to replacing the 11-dimensional worldvolume with $\mathbb{R}^2_{\varepsilon_1} \times \frac{\mathbb{R}^2_{\varepsilon_2} \times \mathbb{R}^2_{\varepsilon_3}}{\mathbb{Z}_K} \times \mathbb{R}_t \times \mathbb{C}_x \times \mathbb{C}_z$. The action of the 5d Chern-Simons theory now involves the $\mathfrak{gl}(K)$-valued gauge field, leading to more complex interaction vertices. Such a study would lead to a direct confirmation of the predictions made in [36].

**Quantum integrability.** We have shown that the intersection of a line defect and a surface defect in the 5d Chern-Simons theory provides an R-matrix of the affine Yangian of $\mathfrak{gl}(1)$. This is reminiscent of the topological line defects in the 4d Chern-Simons theory [20, 24, 41], where their intersections are joined together to construct an integrable model defined on the representations of quantum algebra assigned to the line defects. Meanwhile, it was suggested in [42] that the twisted M-theory is dual to the IIB theory for the gauge origami configuration [43, 44], up to a certain unrefinement of the $\Omega$-background. Through this duality, the M-branes can be introduced to engineer the 4d $\mathcal{N} = 2$ gauge theories and *qq*-characters [43] in the dual side. It is a work in progress to directly connect the quantum integrability emerging from the 4d $\mathcal{N} = 2$ theory to the 5d Chern-Simons theory that we have studied so far. For the study of the former, see [42, 45–59].

# A    Affine Yangian of $\mathfrak{gl}(1)$

## A.1    Presentation

The affine Yangian of $\mathfrak{gl}(1)$ is an associative algebra with generators $e_i$, $f_i$, and $\psi_i$, $i \in \mathbb{Z}_{\geq 0}$ with relations

$$0 = [\psi_i, \psi_j] \tag{A.1a}$$

$$0 = [e_{i+3}, e_j] - 3[e_{i+2}, e_{j+1}] + 3[e_{i+1}, e_{j+2}] - [e_i, e_{j+3}]$$
$$+ \sigma_2[e_{i+1}, e_j] - \sigma_2[e_i, e_{j+1}] - \sigma_3\{e_i, e_j\} \tag{A.1b}$$

$$0 = [f_{i+3}, f_j] - 3[f_{i+2}, f_{j+1}] + 3[f_{i+1}, f_{j+2}] - [f_i, f_{j+3}]$$
$$+ \sigma_2[f_{i+1}, f_j] - \sigma_2[f_i, f_{j+1}] + \sigma_3\{f_i, f_j\} \tag{A.1c}$$

$$0 = [e_i, f_j] - \psi_{i+j} \tag{A.1d}$$

$$0 = [\psi_{i+3}, e_j] - 3[\psi_{i+2}, e_{j+1}] + 3[\psi_{i+1}, e_{j+2}] - [\psi_i, e_{j+3}]$$
$$+ \sigma_2[\psi_{i+1}, e_j] - \sigma_2[\psi_i, e_{j+1}] - \sigma_3\{\psi_i, e_j\} \tag{A.1e}$$

$$0 = [\psi_{i+3}, f_j] - 3[\psi_{i+2}, f_{j+1}] + 3[\psi_{i+1}, f_{j+2}] - [\psi_i, f_{j+3}]$$
$$+ \sigma_2[\psi_{i+1}, f_j] - \sigma_2[\psi_i, f_{j+1}] + \sigma_3\{\psi_i, f_j\} \tag{A.1f}$$

for $i, j \in \mathbb{Z}_{\geq 0}$ and the boundary relations

$$[\psi_0, e_i] = 0, \qquad [\psi_1, e_i] = 0, \qquad [\psi_2, e_i] = 2e_i \tag{A.2}$$

$$[\psi_0, f_i] = 0, \qquad [\psi_1, f_i] = 0, \qquad [\psi_2, f_i] = -2f_i \tag{A.3}$$

and a generalization of Serre relations

$$0 = \text{Sym}_{i_1, i_2, i_3}[e_{i_1}, [e_{i_2}, e_{i_3+1}]] \tag{A.4}$$

$$0 = \text{Sym}_{i_1, i_2, i_3}[f_{i_1}, [f_{i_2}, f_{i_3+1}]], \tag{A.5}$$

where Sym is the symmetrization over all the indices.

The algebra is parameterized by three complex numbers $\varepsilon_1$, $\varepsilon_2$, and $\varepsilon_3$ constrained by

$$\sigma_1 \equiv \varepsilon_1 + \varepsilon_2 + \varepsilon_3 = 0. \tag{A.6}$$

They appear symmetrically through

$$\sigma_2 \equiv \varepsilon_1\varepsilon_2 + \varepsilon_1\varepsilon_3 + \varepsilon_2\varepsilon_3, \qquad \sigma_3 \equiv \varepsilon_1\varepsilon_2\varepsilon_3. \tag{A.7}$$

Let us denote the affine Yangian of $\mathfrak{gl}(1)$ by $Y(\widehat{\mathfrak{gl}}(1))$.

We define the generating functions as formal series in the spectral parameter $x$,

$$e(x) = \sum_{n=0}^{\infty} e_n x^{-n-1}, \quad f(x) = \sum_{n=0}^{\infty} f_n x^{-n-1}, \quad \psi(x) = 1 + \sigma_3 \sum_{n=0}^{\infty} \psi_n x^{-n-1}. \tag{A.8}$$

In terms of the generating functions, the commutation relations can be written as

$$
\begin{aligned}
e(x)e(x') &\sim \varphi(x - x')e(x')e(x) \\
f(x)f(x') &\sim \varphi(x' - x)f(x')f(x) \\
\psi(x)e(x') &\sim \varphi(x - x')e(x')\psi(x) \\
\psi(x)f(x') &\sim \varphi(x' - x)f(x')\psi(x) \\
[e(x), f(x')] &= -\frac{1}{\sigma_3}\frac{\psi(x) - \psi(x')}{x - x'},
\end{aligned}
\tag{A.9}
$$

where

$$
\varphi(x) = \prod_{a=1}^{3}\frac{x + \varepsilon_a}{x - \varepsilon_a}.
\tag{A.10}
$$

Here, note that the relations (A.9) are not equalities because they are not valid for the lowest-order terms. The Serre relations can be written as

$$
\begin{aligned}
0 &= \sum_{\pi \in S_3}\left(x_{\pi(1)} + 2x_{\pi(2)} - x_{\pi(3)}\right)e(x_{\pi(1)})e(x_{\pi(2)})e(x_{\pi(3)}) \\
0 &= \sum_{\pi \in S_3}\left(x_{\pi(1)} + 2x_{\pi(2)} - x_{\pi(3)}\right)f(x_{\pi(1)})f(x_{\pi(2)})f(x_{\pi(3)}).
\end{aligned}
\tag{A.11}
$$

## A.2 Coproduct

The (formal) coproduct $\Delta : Y(\widehat{\mathfrak{gl}}(1)) \to Y(\widehat{\mathfrak{gl}}(1))\widehat{\otimes}Y(\widehat{\mathfrak{gl}}(1))$ of the affine Yangian of $\mathfrak{gl}(1)$ is defined by

$$
\begin{aligned}
\Delta(e_0) &= e_0 \otimes 1 + 1 \otimes e_0 \\
\Delta(f_0) &= f_0 \otimes 1 + 1 \otimes f_0 \\
\Delta(\psi_3) &= \psi_3 \otimes 1 + 1 \otimes \psi_3 + \sigma_3(\psi_2 \otimes \psi_0 + \psi_1 \otimes \psi_1 + \psi_0 \otimes \psi_2) + 6\sigma_3 \sum_{m>0} mJ_m \otimes J_{-m}.
\end{aligned}
\tag{A.12}
$$

Here $J_m = -\frac{1}{(m-1)!}\mathrm{ad}_{f_1}^{m-1}f_0$ and $J_{-m} = \frac{1}{(m-1)!}\mathrm{ad}_{e_1}^{m-1}e_0$ for $m > 0$. By requiring $\Delta$ to be an algebra morphism, we can extend it to all the other generators. For instance,

$$
\begin{aligned}
\Delta(e_1) &= e_1 \otimes 1 + 1 \otimes e_1 + \sigma_3\psi_0 \otimes e_0 \\
\Delta(f_1) &= f_1 \otimes 1 + 1 \otimes f_1 + \sigma_3 f_0 \otimes \psi_0 \\
\Delta(\psi_0) &= \psi_0 \otimes 1 + 1 \otimes \psi_0 \\
\Delta(\psi_1) &= \psi_1 \otimes 1 + 1 \otimes \psi_1 + \sigma_3\psi_0 \otimes \psi_0 \\
\Delta(\psi_2) &= \psi_2 \otimes 1 + 1 \otimes \psi_2 + \sigma_3\psi_1 \otimes \psi_0 + \sigma_3\psi_0 \otimes \psi_1.
\end{aligned}
\tag{A.13}
$$

When restricted to $Y(\widehat{\mathfrak{gl}}(1))/(\psi_0)$, in which the generators are repackaged into $(t_{m,n})_{m\geq 0, n\in\mathbb{Z}}$, the coproduct is written as

$$
\begin{aligned}
\Delta(t_{0,n}) &= t_{0,n} \otimes 1 + 1 \otimes t_{0,n} \\
\Delta(t_{2,0}) &= t_{2,0} \otimes 1 + 1 \otimes t_{2,0} + 2\sigma_3 \sum_{n=0}^{\infty}(n+1)(t_{0,n} \otimes t_{0,-n-2}).
\end{aligned}
\tag{A.14}
$$

When restricted to $Y_1(\widehat{\mathfrak{gl}}(1))$, it becomes a mixed coproduct $\Delta : Y_1(\widehat{\mathfrak{gl}}(1)) \to Y_1(\widehat{\mathfrak{gl}}(1)) \widehat{\otimes} \left( Y(\widehat{\mathfrak{gl}}(1))/(\psi_0) \right)$.

Let us define a formal shift map $S(z) : Y(\widehat{\mathfrak{gl}}(1))/(\psi_0) \to Y_1(\widehat{\mathfrak{gl}}(1))((z))$ by [36]

$$
\begin{aligned}
S(z)(t_{2,0}) &= t_{2,0} \\
S(z)(t_{0,n}) &= \sum_{m=0}^{n} \binom{n}{m} z^{n-m} t_{0,m}, \qquad n \geq 0 \\
S(z)(t_{0,-n}) &= \sum_{m=0}^{\infty} \frac{(-1)^m (n+m-1)!}{m!(n-1)!} z^{-n-m} t_{0,m}, \qquad n \geq 1.
\end{aligned}
\tag{A.15}
$$

By applying $\mathrm{id} \otimes S(z)$ to the coproduct (A.14), we obtain a mixed meromorphic coproduct $\Delta(z) : Y(\widehat{\mathfrak{gl}}(1))/(\psi_0) \to \left( Y(\widehat{\mathfrak{gl}}(1))/(\psi_0) \right) \widehat{\otimes} Y_1(\widehat{\mathfrak{gl}}(1))((z))$, given by

$$
\begin{aligned}
\Delta(z)(t_{0,n}) &= t_{0,n} \otimes 1 + 1 \otimes \sum_{m=0}^{n} \binom{n}{m} z^{n-m} t_{0,m}, \qquad n \geq 0, \\
\Delta(z)(t_{0,-n}) &= t_{0,-n} \otimes 1 + 1 \otimes \sum_{m=0}^{\infty} \frac{(-1)^m (n+m-1)!}{m!(n-1)!} z^{-n-m} t_{0,m}, \qquad n \geq 1, \\
\Delta(z)(t_{2,0}) &= t_{2,0} \otimes 1 + 1 \otimes t_{2,0} + 2\sigma_3 \sum_{m,n \geq 0} \frac{(m+n+1)!}{m!n!} (-1)^n z^{-m-n-2} t_{0,m} \otimes t_{0,n}.
\end{aligned}
\tag{A.16}
$$

## A.3 Vector representations

For any given spectral parameter $\sigma \in \mathbb{C}$, consider the vector space $\mathcal{V}_1(\sigma) = \bigoplus_{i \in \mathbb{Z}} \mathbb{C}[\sigma]_i$. The vector representation of the affine Yangian of $\mathfrak{gl}(1)$ on this space is defined by [60]

$$
\begin{aligned}
e(x)[\sigma]_i &= \frac{1}{\varepsilon_1} \left( \frac{1}{x - (\sigma + i\varepsilon_1)} \right)^+ [\sigma]_{i+1} \\
f(x)[\sigma]_i &= -\frac{1}{\varepsilon_1} \left( \frac{1}{x - (\sigma + (i-1)\varepsilon_1)} \right)^+ [\sigma]_{i-1} \\
\psi(x)[\sigma]_i &= \left( \frac{(x - (\sigma + i\varepsilon_1 + \varepsilon_2))(x - (\sigma + i\varepsilon_1 + \varepsilon_3))}{(x - (\sigma + i\varepsilon_1))(x - (\sigma + (i-1)\varepsilon_1))} \right)^+ [\sigma]_i
\end{aligned}
\tag{A.17}
$$

where $(\cdots)^+$ denotes the formal expansion in large $x$. Note that its level is $(\psi_0, \psi_1) = (0, \varepsilon_1^{-1})$. There are two other vector representations $\mathcal{V}_2(\sigma)$ and $\mathcal{V}_3(\sigma)$, obtained by cyclic permutations of $(\varepsilon_1, \varepsilon_2, \varepsilon_3)$.

Identifying $[\sigma]_i$ with $z^{-i}$ where $z$ is the coordinate on $\mathbb{C}^*$, we can explicitly write down

image of a generating set of $Y(\widehat{\mathfrak{gl}}(1))$ as differential operators on $\mathbb{C}^*$:

$$
\begin{aligned}
\frac{1}{(n-1)!} \mathrm{ad}_{e_1}^{n-1} e_0 &\mapsto \frac{1}{\varepsilon_1} z^{-n} \\
-\frac{1}{(n-1)!} \mathrm{ad}_{f_1}^{n-1} f_0 &\mapsto \frac{1}{\varepsilon_1} z^n \\
\psi_1 &\mapsto \frac{1}{\varepsilon_1} \\
e_1 &\mapsto \frac{\sigma}{\varepsilon_1} z^{-1} - \partial_z \\
[e_2, e_1] &\mapsto \frac{1}{\varepsilon_1} \left( \sigma z^{-1} - \varepsilon_1 \partial_z \right)^2 .
\end{aligned}
\qquad n \geq 1
\tag{A.18}
$$

The M2 brane representation $\mathrm{M2}_{1,0,0}$ is identified with $\mathcal{V}_1(0)$. More generally, the representation $\mathrm{M2}_{N,0,0}$ is a formally a complete tensor product of $N$-copies of $\mathcal{V}_1(0)$ which is induced by the coproduct (A.12). More precisely, every element $g \in Y(\widehat{\mathfrak{gl}}(1))$ acts on $\mathcal{V}_1(0) \widehat{\otimes} \cdots \widehat{\otimes} \mathcal{V}_1(0)$ as $\Delta^{N-1}(g)$, which is written as a formal power series of differential operators in $N$-variables $(x_1, \cdots, x_N)$. One finds that the formal power series actually re-summed to a rational function. For instance, we can explicitly write down image of a generating set of $Y(\widehat{\mathfrak{gl}}(1))$ as differential operators on $\mathbb{C}^{*N}$:

$$
\begin{aligned}
\frac{1}{(n-1)!} \mathrm{ad}_{e_1}^{n-1} e_0 &\mapsto \frac{1}{\varepsilon_1} \sum_{i=1}^N z_i^{-n} \\
-\frac{1}{(n-1)!} \mathrm{ad}_{f_1}^{n-1} f_0 &\mapsto \frac{1}{\varepsilon_1} \sum_{i=1}^N z_i^n \\
\psi_1 &\mapsto \frac{N}{\varepsilon_1} \\
e_1 &\mapsto -\sum_{i=1}^N \partial_{z_i} \\
[e_2, e_1] &\mapsto \varepsilon_1 \sum_{i=1}^N \partial_{z_i}^2 + \frac{\sigma_3}{\varepsilon_1^2} \sum_{i<j}^N \frac{2}{(z_i - z_j)^2} .
\end{aligned}
\qquad n \geq 1
\tag{A.19}
$$

## A.4 Shift automorphisms

A shift in the arguments of the generating currents

$$
\tau_\sigma : e(x), f(x), \psi(x) \mapsto e(x - \sigma), f(x - \sigma), \psi(x - \sigma)
\tag{A.20}
$$

defines a shift automorphism of $Y(\widehat{\mathfrak{gl}}(1))$.

It is easy to see from (A.17) that the pullback of vector representation along $\tau_\sigma$ shifts the spectral parameter by $\sigma$, namely:

$$
\tau_\sigma^* \mathcal{V}_1(\sigma') = \mathcal{V}_1(\sigma' + \sigma).
\tag{A.21}
$$

From the image of a generating set of $Y(\widehat{\mathfrak{gl}}(1))$ in (A.18), one can read out that the effect of shifting spectral parameter $\mathcal{V}_1(\sigma') \mapsto \mathcal{V}_1(\sigma' + \sigma)$ is equivalent to conjugation action by $z^{\delta/\varepsilon_1}$ on the ring of differential operators on $\mathbb{C}^*$. Namely, the following diagram commutes:

$$
\begin{array}{ccc}
Y(\widehat{\mathfrak{gl}}(1)) & \longrightarrow & D(\mathbb{C}^*) \\
\downarrow{\scriptstyle \tau_\sigma} & & \downarrow{\scriptstyle z^{\delta/\varepsilon_1}(\cdots)z^{-\delta/\varepsilon_1}} \\
Y(\widehat{\mathfrak{gl}}(1)) & \longrightarrow & D(\mathbb{C}^*)
\end{array}
\tag{A.22}
$$

where the horizontal arrows are homomorphism from $Y(\widehat{\mathfrak{gl}}(1))$ to ring of differential operators on $\mathbb{C}^*$ induced by $\mathcal{V}_1(0)$.

More generally, let us consider the representation $\mathrm{M2}_{N,0,0}$ which a formally a complete tensor product of $N$-copies of $\mathcal{V}_1(0)$. $\mathrm{M2}_{N,0,0}$ is induced by a homomorphism $Y(\widehat{\mathfrak{gl}}(1)) \to D(\mathbb{C}^{*N})$ which is uniquely determined by the image for a set of generators given by (A.19). We notice that the shift automorphism is compatible with coproduct:

$$
\Delta \circ \tau_\sigma = (\tau_\sigma \otimes \tau_\sigma) \circ \Delta.
\tag{A.23}
$$

Therefore it follows from (A.22) that we have the following commutativity:

$$
\begin{array}{ccc}
Y(\widehat{\mathfrak{gl}}(1)) & \longrightarrow & D(\mathbb{C}^{*N}) \\
\downarrow{\scriptstyle \tau_\sigma} & & \downarrow{\scriptstyle \mathrm{Ad}((z_1\cdots z_N)^{\delta/\varepsilon_1})} \\
Y(\widehat{\mathfrak{gl}}(1)) & \longrightarrow & D(\mathbb{C}^{*N})
\end{array}
\tag{A.24}
$$

Here $\mathrm{Ad}(g)$ means the adjoint action $g(\cdot)g^{-1}$.

The shift automorphism that we used in (5.5) is the conjugation action by $\exp(-\delta t_{0,1})$, which cannot be in the form of $\tau_\sigma$ for any $\sigma$, as we can see from the commutative diagram:

$$
\begin{array}{ccc}
Y(\widehat{\mathfrak{gl}}(1)) & \longrightarrow & D(\mathbb{C}^{*N}) \\
{\scriptstyle \mathrm{Ad}(\exp(-\delta t_{0,1}))}\downarrow & & \downarrow{\scriptstyle \mathrm{Ad}\left(\exp\left(-\frac{\delta}{\varepsilon_1}\sum_i z_i\right)\right)} \\
Y(\widehat{\mathfrak{gl}}(1)) & \longrightarrow & D(\mathbb{C}^{*N})
\end{array}
\tag{A.25}
$$

## A.5 Isomorphism between $Y(\widehat{\mathfrak{gl}}(1))$ and mode algebra of $\mathcal{W}_\infty$

Let us denote the mode algebra of $\mathcal{W}_\infty$ by $U(\mathcal{W}_\infty)$. The isomorphism $Y(\widehat{\mathfrak{gl}}(1)) \xrightarrow{\sim} U(\mathcal{W}_\infty)$ is generated by

$$
\begin{aligned}
\frac{1}{(n-1)!}\mathrm{ad}_{e_1}^{n-1}e_0 &\mapsto J_{-n} \\
-\frac{1}{(n-1)!}\mathrm{ad}_{f_1}^{n-1}f_0 &\mapsto J_n \qquad\qquad n \geq 1 \\
\psi_1 &\mapsto J_0 \\
\psi_3 &\mapsto V_0 - \sigma_3\psi_0 L_0 - \frac{1}{2}\sigma_3 J_0^2 - \frac{3}{2}\sigma_3 \sum_{m\in\mathbb{Z}}|m| : J_{-m}J_m :,
\end{aligned}
\tag{A.26}
$$

where $V_0$ is the zero mode of the quasi-primary field

$$V(z) = W_3(z) + \frac{2}{\psi_0} : J(z)T(z) : - \frac{2}{3\psi_0^2} : J(z)J(z)J(z) : . \tag{A.27}$$

We get, for instance,

$$e_1 \mapsto L_{-1}, \quad \psi_2 \mapsto 2L_0, \quad f_1 \mapsto -L_1,$$
$$\frac{1}{2}[e_2, e_0] - \frac{\sigma_3 \psi_0}{2}[e_1, e_0] \mapsto L_{-2}, \quad -\frac{1}{2}[f_2, f_0] + \frac{\sigma_3 \psi_0}{2}[f_1, f_0] \mapsto L_2. \tag{A.28}$$

From the $[L_2, L_{-2}] = 4L_0 + \frac{1}{2}c_{1+\infty}$ commutator, we get the central charge $c_{1+\infty}$ of $\mathcal{W}_\infty$ expressed in terms of the affine Yangian generators as

$$-\sigma_2 \psi_0 - \sigma_3^2 \psi_0^3 \mapsto c_{1+\infty} = 1 + (\lambda_1 - 1)(\lambda_2 - 1)(\lambda_3 - 1), \tag{A.29}$$

where $\mathcal{W}_\infty$ algebra is parameterized by $(\lambda_1, \lambda_2, \lambda_3)$ satisfying

$$0 = \frac{1}{\lambda_1} + \frac{1}{\lambda_2} + \frac{1}{\lambda_3}. \tag{A.30}$$

From (A.29), we get the map between parameters

$$-\psi_0 \frac{\sigma_3}{\varepsilon_a} \mapsto \lambda_a, \qquad a = 1, 2, 3. \tag{A.31}$$

For each triple of non-negative integers $(L, M, N)$, the $\mathcal{W}_\infty$ algebra has a proper ideal $\mathcal{I}_{L,M,N}$ generated by a singular vector at level $(L+1)(M+1)(N+1)$, if the parameters are further constrained by

$$1 = \frac{L}{\lambda_1} + \frac{M}{\lambda_2} + \frac{N}{\lambda_3}. \tag{A.32}$$

At these specialized parameters, we can take the quotient of the $\mathcal{W}_\infty$ algebra by the proper ideal $\mathcal{I}_{L,M,N}$, called the $Y_{L,M,N}$-algebra; namely,

$$Y_{L,M,N} = \mathcal{W}_\infty / \mathcal{I}_{L,M,N}. \tag{A.33}$$

In terms of the affine Yangian parameters, the constraint (A.32) amounts to set

$$\lambda_a = \frac{L\varepsilon_1 + M\varepsilon_2 + N\varepsilon_3}{\varepsilon_a}, \qquad a = 1, 2, 3, \tag{A.34}$$

so that

$$\psi_0 = -\frac{L\varepsilon_1 + M\varepsilon_2 + N\varepsilon_3}{\sigma_3}. \tag{A.35}$$

The isomorphism between $Y(\widehat{\mathfrak{gl}}(1))$ and $U(\mathcal{W}_\infty)$ implies that there exists an algebra coproduct $\Delta : U(\mathcal{W}_\infty) \to U(\mathcal{W}_\infty) \widehat{\otimes} U(\mathcal{W}_\infty)$. In fact, the aforementioned coproduct is induced from vertex algebra homomorphism $\Delta_\mathcal{W} : \mathcal{W}_\infty \to \mathcal{W}_\infty \otimes \mathcal{W}_\infty$. $\Delta_\mathcal{W}$ is the uniform-in-$N$ version

of the vertex algebra map $\Delta_{N_1,N_2} : \mathcal{W}_{N_1+N_2} \to \mathcal{W}_{N_1} \otimes \mathcal{W}_{N_2}$ which comes from splitting Miura operator into two:

$$(\varepsilon_1 \partial_z - \varepsilon_2 \varepsilon_3 J_1(z)) \cdots (\varepsilon_1 \partial_z - \varepsilon_2 \varepsilon_3 J_{N_1+N_2}(z)) \mapsto$$
$$(\varepsilon_1 \partial_z - \varepsilon_2 \varepsilon_3 J_1(z)) \cdots (\varepsilon_1 \partial_z - \varepsilon_2 \varepsilon_3 J_{N_1}(z)) \times (\varepsilon_1 \partial_z - \varepsilon_2 \varepsilon_3 J_{N_1+1}(z)) \cdots (\varepsilon_1 \partial_z - \varepsilon_2 \varepsilon_3 J_{N_1+N_2}(z)), \tag{A.36}$$

In the $U$-basis, $\Delta_{N_1,N_2}$ is explicitly given by:

$$\Delta_{N_1,N_2}(U_n(z)) = \sum_{\substack{s,t,u \geq 0 \\ s+t+u=n}} \binom{N_1 - t}{u} U_t(z) \otimes \epsilon_1^u \partial^u U_s(z), \tag{A.37}$$

where we set $U_0(z) = 1$. The uniform-in-$N$ version of (A.37) is

$$\Delta_{\mathcal{W}}(U_n(z)) = \sum_{\substack{s,t,u \geq 0 \\ s+t+u=n}} \binom{\lambda_1 \otimes 1 - t}{u} U_t(z) \otimes \epsilon_1^u \partial^u U_s(z), \tag{A.38}$$
$$\Delta_{\mathcal{W}}(\lambda_1) = \lambda_1 \otimes 1 + 1 \otimes \lambda_1.$$

Here $\binom{\lambda_1 \otimes 1 - t}{u}$ means $\frac{1}{u!} \prod_{i=0}^{u-1} (\lambda_1 \otimes 1 - t - i)$ if $u \neq 0$ and it is set to be 1 if $u = 0$.

## A.6 Primary basis of $\mathcal{W}_\infty$

The OPEs of the spin-1 and spin-2 currents in the primary basis are

$$J(z)J(w) \sim \frac{\psi_0}{(z-w)^2}$$
$$T(z)J(w) \sim \frac{J(w)}{(z-w)^2} + \frac{\partial J(w)}{z-w} \tag{A.39}$$
$$T(z)T(w) \sim -\frac{\sigma_2 \psi_0 + \sigma_3^2 \psi_0^3}{2} \frac{1}{(z-w)^4} + \frac{2T(w)}{(z-w)^2} + \frac{\partial T(w)}{z-w}.$$

The $\mathcal{W}_\infty$-algebra is generated by $J$, $T$, and a degree 3 primary field $W^{(3)}$. The higher-spin currents in $\mathcal{W}_\infty$ are not uniquely fixed by requiring them to be primary, even at spin-3 [6]. For instance the following spin-3 field is primary:

$$6\psi_0 : J(z)T(z) : + \left(\sigma_2 \psi_0 + \sigma_3^2 \psi_0^3 - 2\right) : J(z)^3 : . \tag{A.40}$$

The coupling of the surface defect in the 5d CS theory is expected to be given in a basis chosen in a certain way, which can in principle be determined by extending our Feynman diagram computations for their OPEs. We leave this exercise to interested readers.

## B  1-shifted affine Yangian of $\mathfrak{gl}(1)$

The 1-shifted affine Yangian of $\mathfrak{gl}(1)$, which we denote by $Y_1(\widehat{\mathfrak{gl}}(1))$, is an associative algebra generated by $t_{m,n}$ $(m,n \in \mathbb{Z}_{\geq 0})$. The basic generators are $t_{2,0}$ and $t_{0,d}$ from which all the other generators are obtained recursively by

$$[t_{2,0}, t_{c,d}] = 2dt_{c+1,d-1}. \tag{B.1}$$

Also, all the commutation relations can be recursively computed by

$$[t_{3,0}, t_{0,d}] = 3dt_{2,d-1} - \sigma_2 \frac{d(d-1)(d-2)}{4} t_{0,d-3} + \frac{3}{2}\sigma_3 \sum_{m=0}^{d-3}(m+1)(d-m-2)t_{0,m}t_{0,d-3-m}, \tag{B.2}$$

where $\sigma_2 = \varepsilon_1\varepsilon_2 + \varepsilon_2\varepsilon_3 + \varepsilon_3\varepsilon_1$ and $\sigma_3 = \varepsilon_1\varepsilon_2\varepsilon_3$. Note the manifest triality invariance.

### B.1  Embedding into affine Yangian of $\mathfrak{gl}(1)$

There is an algebra embedding $\iota : Y_1(\widehat{\mathfrak{gl}}(1)) \hookrightarrow Y(\widehat{\mathfrak{gl}}(1))$ given by

$$\begin{aligned} t_{0,n} &\mapsto -\frac{1}{(n-1)!}\mathrm{ad}_{f_1}^{n-1}f_0, \qquad n > 0 \\ t_{2,0} &\mapsto -[e_1, e_2] \\ t_{0,0} &\mapsto \psi_1. \end{aligned} \tag{B.3}$$

It is immediate to see, for instance,

$$2t_{1,0} = [t_{2,0}, t_{0,1}] \mapsto -[[e_2, e_1], f_0] = -[\psi_2, e_1] = -2e_1 \implies t_{1,0} \mapsto -e_1. \tag{B.4}$$

The embedding is more manifest if we write the generators of $Y(\widehat{\mathfrak{gl}}(1))/(\psi_0)$ as $(t_{m,n})_{m\geq 0, n\in\mathbb{Z}}$ by including

$$t_{0,-n} = \frac{1}{(n-1)!}\mathrm{ad}_{e_1}^{n-1}e_0. \tag{B.5}$$

### B.2  Meromorphic coproduct

There is a one-parameter $(z \in \mathbb{C})$ family of meromorphic coproducts $\Delta(z) : Y_1(\widehat{\mathfrak{gl}}(1)) \to Y_1(\widehat{\mathfrak{gl}}(1)) \widehat{\otimes} Y_1(\widehat{\mathfrak{gl}}(1))((z))$, defined by

$$\begin{aligned} \Delta(z)(t_{0,n}) &= t_{0,n} \otimes 1 + 1 \otimes \sum_{m=0}^{n}\binom{n}{m}z^{n-m}t_{0,m}, \\ \Delta(z)(t_{2,0}) &= t_{2,0} \otimes 1 + 1 \otimes t_{2,0} + 2\sigma_3 \sum_{m,n=0}^{\infty}\frac{(m+n+1)!}{m!n!}(-1)^n z^{-m-n-2}t_{0,m} \otimes t_{0,n}. \end{aligned} \tag{B.6}$$

This one-parameter family of meromorphic coproducts is precisely the restriction of the mixed meromorphic coproduct (A.16). In particular, observe that the coproduct (A.14) restricted to $Y_1(\widehat{\mathfrak{gl}}(1))$ becomes a mixed coproduct $\Delta : Y_1(\widehat{\mathfrak{gl}}(1)) \to Y_1(\widehat{\mathfrak{gl}}(1)) \otimes \left(Y(\widehat{\mathfrak{gl}}(1))/(\psi_0)\right)$. Further applying $\mathrm{id} \otimes S(z)$, we recover the meromorphic coproduct $\Delta(z) : Y_1(\widehat{\mathfrak{gl}}(1)) \to Y_1(\widehat{\mathfrak{gl}}(1)) \widehat{\otimes} Y_1(\widehat{\mathfrak{gl}}(1))((z))$ of the 1-shifted affine Yangian of $\mathfrak{gl}(1)$ (B.6).

## C   Vanishing subdiagrams

Here we show that any diagram containing certain subdiagrams vanishes because the subdiagrams themselves vanish identically.

**Subdiagram 1**

The first one:

$$v_1 \quad \text{(figure)} \quad v_2 \quad = \quad 0 , \tag{C.1}$$

regardless of the degree of the interaction (e.g. 1 (2.18a) or 3 (2.18b)) or what else is attached to any of the interaction vertices. This is simply because the wedge product of the above two propagators is a 4-form, whereas the propagators contain the same three unique 1-forms, namely $dt_{12}, d\bar{x}_{12}$, and $d\bar{z}_{12}$ (see 2.14 and 2.15).

**Subdiagram 2**

Next,

$$= \quad 0. \tag{C.2}$$

To see this, we need to look at its evaluation. Two of the three fields at the interaction vertex are contracted, let us assume that the remaining field is uncontracted. Let us also assume that the defect is located at $t = x = 0$. We shall use the coordinates $v_0 = (t, x, \bar{x}, z_0, \bar{z}_0)$, $v_1 = (0, 0, 0, z_1, \bar{z}_1)$, and $v_2 = (0, 0, 0, z_2, \bar{z}_2)$ to refer to the coordinates of the interaction vertex, the operator $W^{(m+1)}$, and the operator $W^{(n+1)}$ respectively. Regardless of which term from which interaction vertex we use for this diagram, there are six ways to contract two of the three fields at the vertex with the two fields coming from the two defect actions. Ignoring overall multiplicative factors, the sum of them is[12]

$$
\int dx \wedge dz_0 \wedge d^2 z_1 \wedge d^2 z_2 \wedge \left( \overbrace{A \wedge \partial_x^\alpha \partial_z^\beta \overbrace{A \wedge \partial_x^\gamma \partial_z^\delta A \partial_x^m A_{\bar{z}_1} \partial_x^n A_{\bar{z}_2}}} \right.
$$

$$
+ \overbrace{A \wedge \partial_x^\alpha \partial_z^\beta A \wedge \partial_x^\gamma \partial_z^\delta A \partial_x^m A_{\bar{z}_1} \partial_x^n A_{\bar{z}_2}} + \overbrace{A \wedge \partial_x^\alpha \partial_z^\beta A \wedge \partial_x^\gamma \partial_z^\delta A \partial_x^m A_{\bar{z}_1} \partial_x^n A_{\bar{z}_2}}
$$

$$
+ \overbrace{A \wedge \partial_x^\alpha \partial_z^\beta A \wedge \partial_x^\gamma \partial_z^\delta A \partial_x^m A_{\bar{z}_1} \partial_x^n A_{\bar{z}_2}} + A \wedge \partial_x^\alpha \partial_z^\beta \overbrace{A \wedge \partial_x^\gamma \partial_z^\delta A \partial_x^m A_{\bar{z}_1} \partial_x^n A_{\bar{z}_2}}
$$

$$
\left. + A \wedge \partial_x^\alpha \partial_z^\beta \overbrace{A \wedge \partial_x^\gamma \partial_z^\delta \overbrace{A \partial_x^m A_{\bar{z}_1} \partial_x^n A_{\bar{z}_2}}} \right) \tag{C.3}
$$

We claim that each of the above *integrands* vanish independently. Take the last term above. To get the correct volume form we need to extract the forms $dt$, $d\bar{x}$, and $d\bar{z}$ from the propagators and the uncontracted field. Since both propagators are forced to have $\bar{z}_1$ and $\bar{z}_2$-components, we can only get the $dt$ and $d\bar{x}$ from the propagators and we must get the $\bar{z}$

---

[12]$\alpha, \beta, \gamma, \delta$ are arbitrary non-negative integers, to allow for arbitrary interaction vertex.

component from the uncontracted field. Collecting the relevant terms (and omitting the forms already written explicitly in the above equation) we find the following expression for the last term:

$$(-1)^{m+n}A_{\bar{z}}\mathrm{d}\bar{z}\wedge\mathrm{d}t\wedge\mathrm{d}\bar{x}\Big(\partial_x^{\alpha+n}\partial_z^{\beta}P_{t\bar{z}}(v_{02})\partial_x^{\gamma+m}\partial_z^{\delta}P_{\bar{x}\bar{z}}(v_{01})-\partial_x^{\alpha+n}\partial_z^{\beta}P_{\bar{x}\bar{z}}(v_{02})\partial_x^{\gamma+m}\partial_z^{\delta}P_{t\bar{z}}(v_{01})\Big). \tag{C.4}$$

Evaluating the expression inside the parentheses we find:

$$\frac{-3\varepsilon_1}{16\pi^2}\prod_{i=0}^{\alpha+\beta+n-1}\left(-\frac{5}{2}-i\right)\frac{\bar{x}^{\alpha+n+1}\bar{z}_{02}^{\beta}}{d_{02}^{\frac{5}{2}+\alpha+\beta+n}}\times\frac{3\varepsilon_1}{32\pi^2}\prod_{j=0}^{\gamma+\delta+m-1}\left(-\frac{5}{2}-j\right)\frac{t\bar{x}^{\gamma+m}\bar{z}_{01}^{\delta}}{d_{01}^{\frac{5}{2}+\gamma+\delta+m}}$$

$$-\frac{3\varepsilon_1}{32\pi^2}\prod_{i=0}^{\alpha+\beta+n-1}\left(-\frac{5}{2}-i\right)\frac{t\bar{x}^{\alpha+n}\bar{z}_{02}^{\beta}}{d_{02}^{\frac{5}{2}+\alpha+\beta+n}}\times\frac{-3\varepsilon_1}{16\pi^2}\prod_{j=0}^{\gamma+\delta+m-1}\left(-\frac{5}{2}-j\right)\frac{\bar{x}^{\gamma+m+1}\bar{z}_{01}^{\delta}}{d_{01}^{\frac{5}{2}+\gamma+\delta+m}} \tag{C.5}$$

$$=0.$$

Using the fact that the uncontracted field plays no role in this computation and relabeling the arbitrary integers $\alpha,\beta,\gamma,\delta,m,n$ as needed or setting some to zero, we notice that, in fact, all six terms in (C.3) vanish by essentially the same computation.

**Subdiagram 3**

Essentially similar computation leads to:

$$= \quad 0. \tag{C.6}$$

**Subdiagram 4**

If we replace one of the propagators in (C.2) with a background field we get another vanishing subdiagram.

$$= \quad 0. \tag{C.7}$$

The proof is this is similar to that of (C.2). However, we provide some details below as it is not a priori obvious.

Since one of the fields at the interaction vertex has been replaced by $A^{(0)}$, there are two ways to contract one of the remaining fields with the field coming from the surface defect, giving us:

$$\int\mathrm{d}x\wedge\mathrm{d}z_0\wedge\mathrm{d}^2z_1\wedge\left(\overline{A\wedge\partial_x^{\alpha}\partial_z^{\beta}A}\wedge\partial_x^{\gamma}A^{(0)}\partial_x^m A_{\bar{z}_1}+A\wedge\partial_x^{\alpha}\partial_z^{\beta}\overline{A\wedge\partial_x^{\gamma}A^{(0)}\partial_x^m A_{\bar{z}_1}}\right). \tag{C.8}$$

There are of course, three ways to choose which field at the interaction vertex is to be replaced by $A^{(0)}$, but the computation will be essentially the same for all of them. Similar to (C.3), here also the $\mathrm{d}\bar{z}_0$ component of the volume form must come from the uncontracted field. Because, from the propagator and the background field we can only get $\mathrm{d}t \wedge \mathrm{d}\bar{x}$. Keeping the relevant components from the last term inside the parentheses above we get

$$(-1)^m A_{\bar{z}_0} \mathrm{d}\bar{z}_0 \wedge \mathrm{d}t \wedge \mathrm{d}\bar{x}\Big( \partial_x^{\alpha+m} \partial_z^\beta P_{t\bar{z}}(v_{01}) \partial_x^{\gamma-1} F_{x\bar{x}}^{(0)}(t,x,\bar{x}) - \partial_x^{\alpha+m} \partial_z^\beta P_{\bar{x}\bar{z}}(v_{01}) \partial_x^{\gamma-1} F_{xt}^{(0)}(t,x,\bar{x}) \Big). \tag{C.9}$$

Evaluating the terms inside the parentheses we get:

$$\frac{-3\varepsilon_1}{16\pi^2} \prod_{i=0}^{\alpha+\beta+m-1} \left( -\frac{5}{2} - i \right) \frac{\bar{x}^{\alpha+m+1} \bar{z}_{01}^\beta}{d_{01}^{\frac{5}{2}+\alpha+\beta+m}} \times \frac{N\varepsilon_1}{4\varepsilon} \prod_{j=0}^{\gamma-2} \left( -\frac{3}{2} - j \right) \frac{t\bar{x}^{\gamma-1}}{(t^2 + |x|^2)^{\frac{1}{2}+\gamma}}$$

$$- \frac{3\varepsilon_1}{32\pi^2} \prod_{i=0}^{\alpha+\beta+m-1} \left( -\frac{5}{2} - i \right) \frac{t\bar{x}^{\alpha+m} \bar{z}_{01}^\beta}{d_{01}^{\frac{5}{2}+\alpha+\beta+m}} \times \frac{-N\varepsilon_1}{2\varepsilon} \prod_{j=0}^{\gamma-2} \left( -\frac{3}{2} - j \right) \frac{\bar{x}^\gamma}{(t^2 + |x|^2)^{\frac{1}{2}+\gamma}} \tag{C.10}$$

$$= 0.$$

**Subdiagram 5**

Similarly we get:

$$= \quad 0. \tag{C.11}$$

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
