# Peer review of "R-matrices and Miura operators in 5d Chern-Simons theory"

_SciPost Physics Core_

## Round 1 · Referee Report · Anonymous (Referee 1) · 2024-9-30

Strengths

  1. This paper fills an important gap in the twisted M-theory & integrability literature. The study of certain W and Y algebras, obtained from intersecting, twisted M2-M5 branes coupled to an Omega-deformed (non-commutative) twisted M-theory background, as introduced by Gaiotto and Rapcak, have been of much recent interest to physicists and mathematicians (with connections to the COHA, instanton moduli spaces, etc.). This paper derives the conjectured form of Miura operators for these algebras from first principles using the twisted M-theory setup.

The Miura operators are used to construct free-field realization of the vertex algebras of interest (here, W and Y algebras). They are of further interest from an integrability point of view because they can be interpreted in this setup as R-matrices of a quantum algebra (the affine Yangian of gl(1)).

  1. The paper is quite well-written and the relevant computations are spelled out in excellent detail, which is valuable as these sorts of computations are not (yet) totally standard or familiar. Thus, the paper may also be useful to physicists studying other supersymmetry-preserving intersecting brane setups from the twisted point of view.

Weaknesses

  1. The paper is at times written in an overly mathematical way, so that it may not be as digestible to a reader interested in adapting the authors' formalism.

  2. The final result is also rather technical, which is not a weakness per se, but may obscure the broader relevance and physical interest of this setup.

Report

This paper is a nice offering in the burgeoning field of twisted string theory/M-theory, and the main result of the paper, solidifying the conjectural R-matrix interpretation of Miura operators, is supported by many impressive calculations.
It would be very interesting, as the authors note in the conclusions, to leverage this result for integrability of, e.g. resulting 4d N=2 gauge theories in the IIB duality frame, as emphasized by Nekrasov.

I am happy to recommend publication for this work, pending some (relatively minor & expository) changes suggested below.

Requested changes

  1. It may be helpful to have a diagram in the introduction, showing the M2-M5 intersection studied in the present work, as well as using the diagram to spell out some of the geometric intuition behind the conjectures that are proven more formally in the main text.

  2. In the second paragraph on page 3 of the intro, the authors briefly summarize their methods to obtain the OPEs of the various defect algebras appearing in the main text. After or around reference 20, it would be appropriate to mention that this method often goes by the name Koszul duality, and to cite some of the works emphasizing this approach.

  3. In the same paragraph, it would be worth adding a footnote to give at least an informal definition of a (completed) tensor product.

  4. In the final paragraph of page 3, the authors have some discussion of D-brane backreaction and citations to earlier work. It may also be worth mentioning (either in that location or in the relevant computations of section 3), that another recent paper computes D-brane backreaction using the same Feynman diagram approach as this paper: https://inspirehep.net/literature/2779492 .

  5. At the beginning of section 2, could there be some additional explanation of why the D6 brane has a constant B-field turned on.

  6. Relatedly, can the authors briefly discuss, perhaps at the beginning of section 2, the relevant supercharges/how many supercharges are preserved by their M2-M5 configuration, which make their conjectured localization possible?

  7. The authors have "at the leading orders" in a few places, which should be replaced with "to leading order".

  8. The "e.g." on page 6 should be "i.e.".

9.At the beginning of section 2.2, the authors briefly mention the form of the generalized Wilson line follows from topological descent. This may be well-known to experts but not to the general reader, so the authors should include some references, and optionally add a few more details in the text explaining why this is so.

  1. In the sentence right before section 3, "will be taken account of" should be changed to "will be taken into account."

  2. In the first paragraph of section 3, add "a local gauge or BRST" * in "absence of * anomaly", for clarity.

  3. In footnote 3, change "along the defect of local functions" to "of local functions along the defect."

  4. "gauge" is misspelled on the bottom of page 12

  5. In footnote 4, the authors mention some errors in reference [38]. Are they able to explicitly correct or at least point to any important erroneous equations that appear in that work? It may be useful for readers that are also studying the similar computations of reference [38].

  6. In section 4.1, the authors should discuss in simple terms why fusion of defects (which naively looks like a 2-to-1 process) is best thought of mathematically as a coproduct rather than a product.

  7. On page 25, the authors should define "opposite coproduct" (which will be familiar to a mathematical reader but maybe not a physicist).

Recommendation

Publish (easily meets expectations and criteria for this Journal; among top 50%)

---

## Round 1 · Referee Report · Anonymous (Referee 2) · 2024-10-14

Strengths

The connection between the integrable models and string/M theory brane configuration has been one of the major subjects. Some years ago, Costello (ref. [14]) proposed that the symmetry in the integrable models (considered by Maulik-Okounkov and Shiffmann-Vasserot) may be realized in M-theory set-up as the 5-dimensional noncommutative Chern-Simons theory. Later, Gaiotto-Rapcak (ref. [15]) proposed that the Miura operator in the definition of W- and Y-algebras appears in the coproduct of the affine Yangian symmetry in the brane set-up by Costello.

This paper provides a very explicit computation that confirms these proposals. In particular, it uses the Feynman diagram technique, which is familiar among physicists. In this respect, it helps people to understand the abstract and mathematical proposals in [14, 15] more concretely.

In section three, the authors derived the algebra induced by the line and surface defects in the perturbative framework. In particular, from the surface defects, they managed to derive U(1) Kac-Moody and Virasoro algebras, which give a part of the W(infinity) algebra. In section four, they examined the coproduct coming from the set-up. The first nontriviality appearing in the Virasoro generators, eq.(4.19b), is explained at the first-order perturbation. Finally, in section five, they derived the Miura operator as the expectation value of the intersection of line and surface defects. These results explain explicitly how the setup in [14] works to derive the affine Yangian or W(infinity) algebra.

Finally, I would like to mention that the paper is logically well-written, and readers with appropriate backgrounds can easily follow it.

Weaknesses

While it may not be the 'weakness' of the paper, I felt the additional explanation of the nonlinearity of the algebras (W(infinity) or affine Yangian) would be useful. Namely, the nontriviality of W(infinity) algebra shows up when we compute the algebra involving currents with spin three and higher. Also, the notion of the coproduct should become more involved once we include such currents. A short explanation of why we do not need to examine them would be convincing for the readers.

Report

The explicit computation provided by this paper will be very helpful in the future study. I support the publication of the paper.

Requested changes

I would appreciate it if the authors added some extra explanations, which I mention in the 'weakness' part of the report.

Recommendation

Publish (meets expectations and criteria for this Journal)

---

## Editorial Decision

resubmitted